# Unusually high thermal conductivity in suspended monolayer MoSi₂N₄

Chengjian He[1,2,4], Chuan Xu[1,2,4], Chen Chen[1,2,4], Jinmeng Tong[1,2], Tianya Zhou[1,2], Su Sun[1,2], Zhibo Liu[1,2], Hui-Ming Cheng [1,2,3] & Wencai Ren [1,2] ✉

Two-dimensional semiconductors with high thermal conductivity and charge carrier mobility are of great importance for next-generation electronic and optoelectronic devices. However, constrained by the long-held Slack's criteria, the reported two-dimensional semiconductors such as monolayers of MoS₂, WS₂, MoSe₂, WSe₂ and black phosphorus suffer from much lower thermal conductivity than silicon (~142 W·m⁻¹·K⁻¹) because of the complex crystal structure, large average atomic mass and relatively weak chemical bonds. Despite the more complex crystal structure, the recently emerging monolayer MoSi₂N₄ semiconductor has been predicted to have high thermal conductivity and charge carrier mobility simultaneously. In this work, using a noncontact optothermal Raman technique, we experimentally measure a high thermal conductivity of ~173 W·m⁻¹·K⁻¹ at room temperature for suspended monolayer MoSi₂N₄ grown by chemical vapor deposition. First-principles calculations reveal that such unusually high thermal conductivity benefits from the high Debye temperature and small Grüneisen parameter of MoSi₂N₄, both of which are strongly dependent on the high Young's modulus induced by the outmost Si-N bilayers. Our study not only establishes monolayer MoSi₂N₄ as a benchmark 2D semiconductor for next-generation electronic and optoelectronic devices, but also provides an insight into the design of 2D materials for efficient heat conduction.

The continued miniaturization and rapid increase in power density of modern electronics make heat dissipation one of the most critical technological challenges[1–3]. The performance of electronic and optoelectronic devices benefits from semiconductors with simultaneously high electron/hole mobility and high thermal conductivity[4,5]. Compared to three-dimensional (3D) bulk semiconductors such as the well-known silicon, two-dimensional (2D) van der Waals (vdW) layered semiconductors have been widely considered to be ideal atomically thin channels that could facilitate continued miniaturization of next-generation electronics because of the dangling-bond-free surface and little mobility variation with decreasing the thickness[6,7].

The long-held criteria for high thermal conductivity (HTC) materials are: (i) low average atomic mass, (ii) strong interatomic bonding, (iii) a simple crystal structure, and (iv) low anharmonicity[8]. According to these criteria established by Slack, only crystals composed of a small number of strongly bonded atoms of light elements in a primitive crystallographic unit cell would have superhigh thermal conductivity. Graphite[9] and hexagonal boron nitride (h-BN)[10,11] are two most widely used prototypical vdW layered HTC materials that satisfy Slack's criteria. Due to the loss of interlayer phonon scattering[12,13], monolayers of graphite and h-BN exhibit much higher thermal conductivity than their bulk counterparts[14,15], reaching a record thermal conductivity of

[1]Shenyang National Laboratory for Materials Science, Institute of Metal Research, Chinese Academy of Sciences, Shenyang 110016, P. R. China. [2]School of Materials Science and Engineering, University of Science and Technology of China, Shenyang 110016, P. R. China. [3]Shenzhen Institute of Advanced Technology, Chinese Academy of Sciences, Shenzhen 518055, P. R. China. [4]These authors contributed equally: Chengjian He, Chuan Xu, Chen Chen. ✉e-mail: wcren@imr.ac.cn

~5300 $W \cdot m^{-1} \cdot K^{-1}$ for graphene[14] and ~751 $W \cdot m^{-1} \cdot K^{-1}$ for monolayer $h$-BN[15], but they are semi-metal and insulator, respectively. In contrast, the known 2D vdW semiconductors such as transition metal dichalcogenides (TMDCs) have a much lower thermal conductivity than silicon (~142 $W \cdot m^{-1} \cdot K^{-1}$)[16], typically less than 100 $W \cdot m^{-1} \cdot K^{-1}$ [17,18], because of their complex crystal structure, large average atomic mass and relatively weak chemical bonds.

$MoSi_2N_4$ is a newly emerging artificial vdW layered 2D semiconductor with a bandgap of ~1.94 eV for monolayer, which has no natural 3D counterparts and is grown by chemical vapor deposition (CVD)[19]. The monolayer of $MoSi_2N_4$ can be viewed as a $MoN_2$ monolayer sandwiched between two outmost Si-N bilayers (Fig. 1a), and such unique structure endows the material with much higher theoretical electron and hole mobilities, optical transmittance, breaking strength and Young's modulus than those of typical 2D semiconductor, monolayer $MoS_2$[19,20]. Furthermore, many other interesting phenomena and properties have been predicted such as valley pseudospin[21], quantum magneto-transport[22], piezoelectricity[23], sliding ferroelectricity[24], strong exciton-phonon coupling[25], second harmonic generation[26], photocatalysis[27] and tunable Schottky barrier height[28]. These remarkable properties of $MoSi_2N_4$ open up possibilities for various applications from electronics, optoelectronics, spintronics to energy storage and conversion. In particular, despite the more complex structure than TMDCs, such high mobility 2D semiconductor has been predicted to simultaneously have high in-plane thermal conductivity ranging from ~224 to 439 $W \cdot m^{-1} \cdot K^{-1}$ at room temperature[27,29,30], which appears to deviate from the Slack's criteria.

In this work, we experimentally measured a thermal conductivity of ~173 $W \cdot m^{-1} \cdot K^{-1}$ at room temperature for suspended monolayer $MoSi_2N_4$ using a noncontact optothermal Raman technique, which has been widely used to measure the thermal transport properties of mono- or few-layered 2D materials[14,15,31,32]. This value is much higher than those of silicon and the known 2D semiconductors including $MoS_2$[17,33,34], $WS_2$[32], $MoSe_2$[33], $WSe_2$[35] and black phosphorus[36]. First-principles calculations reveal that such unusually high thermal conductivity benefits from the high Debye temperature and low Grüneisen parameter of $MoSi_2N_4$, both of which are strongly dependent on its high Young's modulus induced by the outmost Si-N bilayers.

## Results

### Thermal conductivity measurements of monolayer $MoSi_2N_4$

We grew $MoSi_2N_4$ crystals on Cu/Mo bilayer substrates by passivating the surface dangling bonds of nonlayered 2D molybdenum nitride with elemental silicon during CVD[19], and then transferred them onto various target substrates (see details in Methods). The obtained $MoSi_2N_4$ crystals have a uniform thickness of ~1.2 nm measured by atomic force microscope (AFM) (Fig. 1b, c), corresponding to monolayers. Moreover, such monolayer triangular flakes are single crystals (Fig. 1d), where N atoms are located at the center of the Mo-Si honeycomb lattice (Fig. 1e and Supplementary Fig. 1). We used these single crystals to reveal the intrinsic thermal conductivity of monolayer $MoSi_2N_4$, which avoid the influence of grain boundaries.

The optical absorption coefficient ($\alpha$) at a specific laser wavelength is a necessary parameter for identifying the thermal conductivity of a material by the optothermal Raman technique. To achieve the value of $\alpha$, we grew a monolayer $MoSi_2N_4$ film by extending the growth time and then transferred it onto polished quartz substrate to measure its optical transmittance spectrum (Fig. 1f). The film shows light red color and its transmittance in the visible band ranges from 92.2% to 99.7% with an average of $97.5 \pm 0.2\%$. Specifically, the transmittance at 532 nm, the laser wavelength used for the measurements of thermal conductivity, was identified as 94.48%. The value of $\alpha$ is equal to 1 minus the value of transmittance, i.e., $\alpha = 5.52\%$, because the reflectance rate of monolayer 2D material can be nearly ignored in analogy with that of graphene[37] and monolayer $h$-BN[15]. The precise thickness ($h$) of monolayer $MoSi_2N_4$ is another parameter needed for extracting the thermal conductivity. As shown in Fig. 1g–i, $MoSi_2N_4$ is a vdW layered crystal with sandwich structure for each layer, where $MoN_2$ monolayer is sandwiched between two Si-N bilayers. The interlayer spacing was measured to be 1.07 nm (Fig. 1g), which is used as the thickness of monolayer $MoSi_2N_4$.

The temperature and laser power-dependent shifts of Raman peaks are the basis for identifying the thermal conductivities of 2D materials by using the optothermal Raman technique[14,31,38,39]. Therefore, we first investigated the shift behaviors of the main Raman modes of monolayer $MoSi_2N_4$ transferred on $SiO_2/Si$ substrate at different temperatures. The 532 nm laser was used to collect the Raman spectra of monolayer $MoSi_2N_4$ because it can resonantly enhance with excitons and stimulate stronger Raman signals than other lasers like 488 nm, 633 nm and 785 nm lasers (Supplementary Fig. 2). As shown in Fig. 2a, monolayer $MoSi_2N_4$ has four main Raman peaks located at ~290 $cm^{-1}$, ~350 $cm^{-1}$, ~632 $cm^{-1}$ and ~695 $cm^{-1}$, which correspond to the vibration modes of Si-N, Mo-Si-N, Mo-N and Si-N, respectively (Fig. 2b). For the sake of description, we named them as $SN_1$, $MSN$, $MN$ and $SN_2$, respectively. Notably, all Raman peaks red shift with increasing the temperature or laser power (Fig. 2c, e), similar to those of other 2D materials. This shift is due to the anharmonic coupling of phonon modes induced "self-energy" shift and thermal expansion caused "volume-change" shift[38].

For the optothermal Raman technique, the extracted thermal conductivity is independent of the Raman peak chosen for the analysis[40]. To accurately measure the thermal conductivity, the most sensitive Raman peak shifting to temperature and laser power is preferred. In addition, the recognition of Raman peak position also affects the accuracy of thermal conductivity analysis. The Raman peak is Lorentz linear in nature, while the measured spectrum is usually the convolution of Lorentz function and Gaussian function due to the instrument influence or sample characteristics, which is known as Voigt function[41]. Thus, we obtained the accurate Raman peak position by Voigt function fitting throughout the analyses (Supplementary Fig. 3). Notably, the $SN_1$ mode shows red shifts of ~6.3 $cm^{-1}$ (Fig. 2d and Supplementary Fig. 4a) and ~3.0 $cm^{-1}$ (Fig. 2f and Supplementary Fig. 5) over the examined temperature (293 - 573 K) and laser power (0.25 - 4 mW) range, respectively. In contrast, the red-shifts of other Raman peaks are less than 2.9 $cm^{-1}$ and 1.4 $cm^{-1}$, respectively (Fig. 2d, f and Supplementary Fig. 4). Thus, we used the $SN_1$ mode at ~290 $cm^{-1}$ as the thermometer for optothermal Raman measurements.

Figure 3a shows the schematic of the optothermal Raman measurements of thermal conductivity of monolayer $MoSi_2N_4$. As reported previously, the $SiO_2/Si$ substrate can substantially influence the thermal transport of the 2D materials on the top of it. In this case, the 2D materials suffer from phonon-defect, phonon-impurity and phonon-boundary scatterings as well as heat dissipation along the out-of-plane direction, which will lead to an inaccurate lattice thermal conductivity[42]. On the contrary, the trench or hole can make 2D materials suspended and force heat to propagate along their in-plane direction, which eliminates the thermal coupling of 2D materials to the substrate and avoids extra out-of-plane heat conductions[14,42]. In the measurements, the incident laser focused on the center of the suspended monolayer $MoSi_2N_4$ to avoid heat dissipation through the substrate. The evaporated 100-nm-thick Au film acted as a heat sink to improve the interfacial thermal conductance and accelerate the thermal transfer of the samples for accurate measurements[43]. Previous studies show that the heat convection under ambient condition can affect the thermal conduction of the sample[44]. Thus, all the thermal conductivity measurements were carried out in vacuum in our experiments to ensure that the heat from the laser only travels along in-plane direction of the sample without out-of-plane convection.

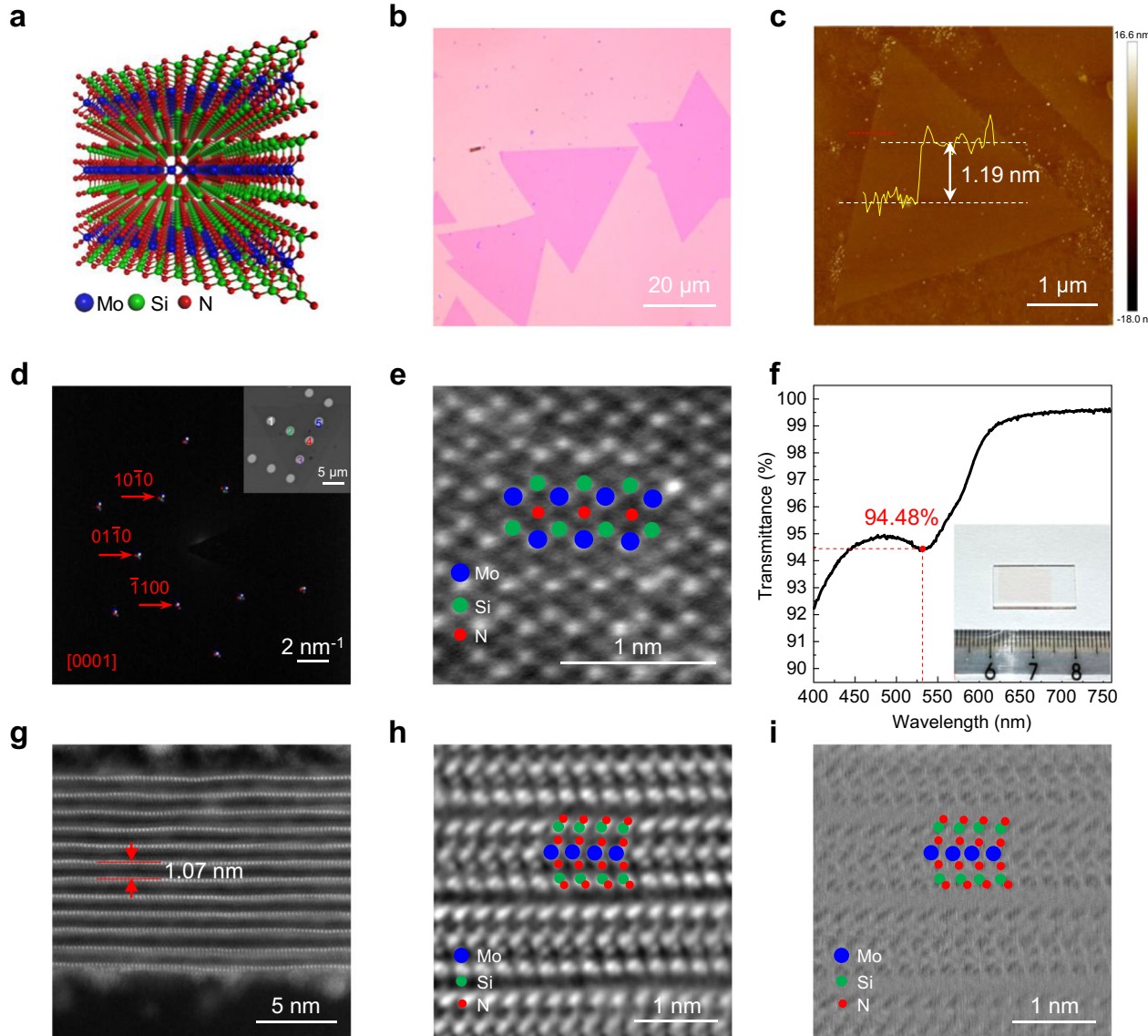

**Fig. 1 | Characterizations of CVD-grown MoSi₂N₄ crystals. a** Atomic model of a trilayer $MoSi_2N_4$. **b, c** Optical (**b**) and AFM (**c**) images of triangular monolayer $MoSi_2N_4$ crystals on $SiO_2$/Si substrate, showing a thickness of ~1.2 nm. **d** Overlapped selective area electron diffraction (SAED) patterns along the [0001] zone axis taken from the 5 positions denoted by colored numbers in the low-magnification transmission electron microscopy (TEM) image (inset). **e** Plan-view integrated differential phase contrast (iDPC)-scanning TEM (STEM) image of monolayer $MoSi_2N_4$ crystal. **f** Transmittance spectrum of a monolayer $MoSi_2N_4$ film transferred onto polished quartz substrate (inset) in visible light band, showing an optical transmittance of 94.48% at 532 nm. **g–i** Atomic-level high-angle annular dark-field (HAADF)- (**g**), iDPC- (**h**), and differentiated differential phase contrast (dDPC)- (**i**) STEM images of the cross section of a multilayer $MoSi_2N_4$ crystal, showing a typical vdW layered structure with an interlayer spacing of 1.07 nm. The blue, green and red balls represent Mo, Si and N atoms, respectively, in **e, h** and **i**.

Figure 3b and c show that the suspended $MoSi_2N_4$ is continuous and smooth without breakage. Raman mapping of the intensity of $SN_1$ mode shows that the Raman signals of the suspended $MoSi_2N_4$ are uniform but much weaker than that of $MoSi_2N_4$ on Au/SiO₂/Si substrate (Fig. 3d, e). Moreover, a red shift of $SN_1$ mode was observed in the suspended region, similar to the suspended graphene[45]. The Raman peaks of a material stem from the vibrations or rotations of its chemical bonds, and the peak position is sensitive to lattice deformation. Generally, a tensile or compressive strain in materials will cause a red shift or blue shift of Raman peaks, respectively, ascribed to the softening or hardening of chemical bonds[46,47]. Different from the $MoSi_2N_4$ supported on Au/SiO₂/Si substrates, there would be a tensile stress in the suspended $MoSi_2N_4$ induced by its own gravity. The corresponding tensile strain results in the red shift of Raman peak.

In addition to $MoSi_2N_4$ itself, the diameter of the hole and laser spot size also affect the heat dissipation of $MoSi_2N_4$. If the hole diameter is too small relative to the laser spot size, the temperature of the sample near the hole edge can't reach ambient temperature (300 K) to meet the heat conduction model. In our experiments, holes of 6 μm in diameter were used, which are much larger than the laser spot, ~1.18 μm in diameter under 50 × telephoto lens. Such hole and laser spot size provide adequate spatial resolution to accurately determine the sample position and to acquire precise Raman spectra at the center of the sample. Importantly, COMSOL simulations indicate that 6-μm-diameter hole and ~1.18-μm-diameter laser spot can make sure the temperature near the hole edge reaches ambient temperature, ensuring the accuracy of thermal conductivity extraction[34,42] (Supplementary Figs. 6–8 and Supplementary Note 1). According to the reported simulation results of monolayer WS₂[32] and MoS₂[34], which have lower thermal conductivity

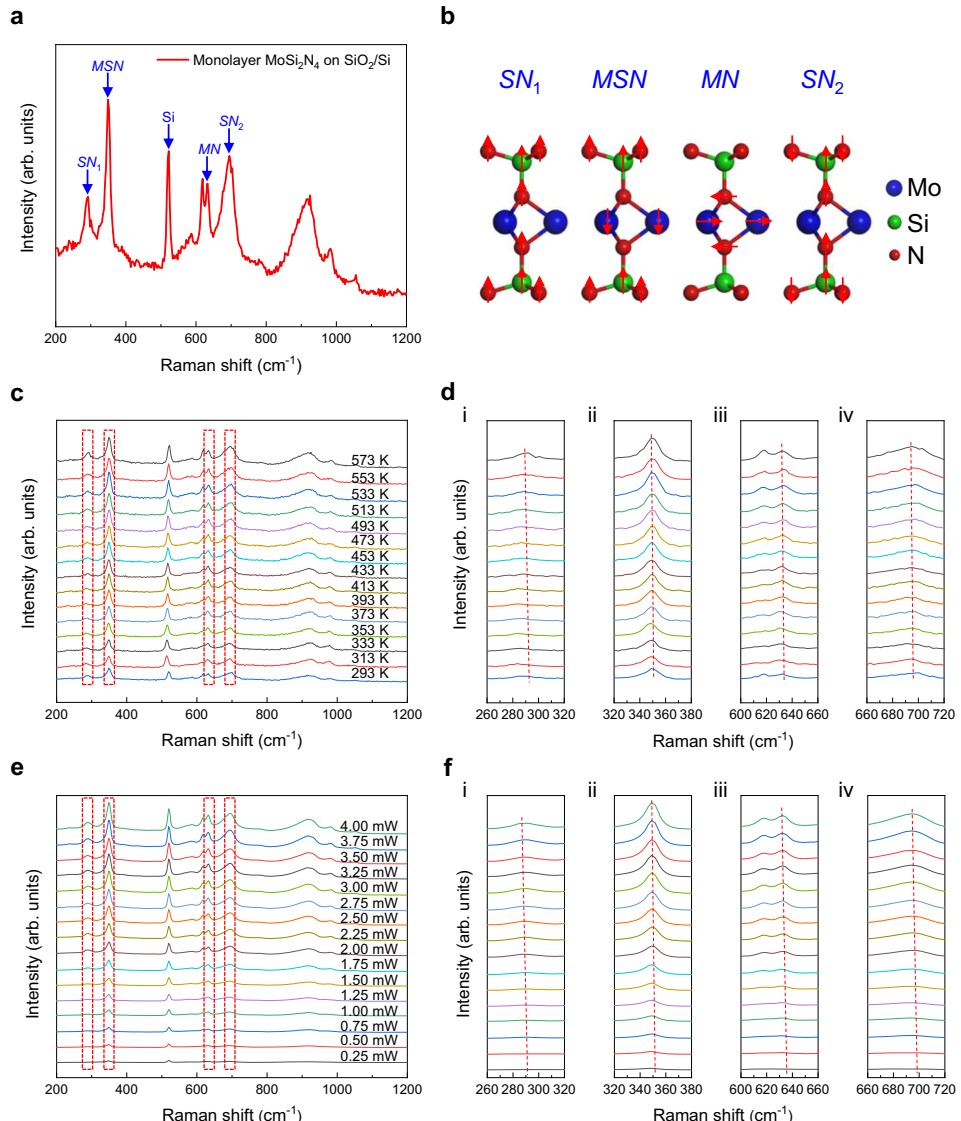

**Fig. 2 | Raman characterizations of monolayer MoSi₂N₄ crystals on SiO₂/Si substrate. a** Typical Raman spectrum under 532 nm laser. **b** Four vibrational modes corresponding to the Raman signals in (**a**). *SN₁*, *MSN*, *MN* and *SN₂* stand for the vibration modes of Si-N, Mo-Si-N, Mo-N and Si-N, respectively. **c**, **e** Raman spectra evolutions with temperature (**c**) and incident laser power (**e**), respectively, where

*SN₁*, *MSN*, *MN* and *SN₂* modes are marked by red dotted boxes from left to right. **d**, **f** The *SN₁* (i), *MSN* (ii), *MN* (iii) and *SN₂* (iv) Raman peak evolutions extracted from **c** and **e**, respectively. The red dotted lines in **d** (i) - (iv) and **f** (i) - (iv) represent the shift trends of Raman modes.

around 30 W·m⁻¹·K⁻¹, 6-µm- and even 1.2-µm-diametered holes are also large enough for the thermal flow transport from the center to the edge.

We collected Raman spectra of suspended monolayer single-crystal MoSi₂N₄ above 6-µm-diametered through-holes by changing temperature or laser power. Figure 3f shows the temperature-dependent Raman spectra from 293 K to 573 K with a step of 20 K. The power of incident laser was kept as low as 200 µW to avoid unexpected laser heating. Figure 3i shows the laser power-dependent Raman spectra over the range of 0.13 mW to 3.00 mW with a step of 0.12 mW and 0.13 mW alternately. With increasing the temperature or laser power, all the Raman peaks of suspended monolayer MoSi₂N₄ show red shifts, which are the same as the shift trends on SiO₂/Si substrate. Figure 3g and j show the dependences of the most sensitive *SN₁* mode on the temperature and laser power, respectively. The first-order temperature coefficient χ of monolayer MoSi₂N₄ can be extracted from the temperature-dependent *SN₁* mode frequency shift

with the equation[15,38]

$$\omega(T) = \omega_0 + \chi \times T \tag{1}$$

where $\omega_0$ is the frequency of the *SN₁* mode when the absolute temperature *T* is extrapolated to 0 K. We extracted the Raman spectrum near the *SN₁* mode (from 260 cm⁻¹ to 320 cm⁻¹) and fitted it with Voigt function to obtain the accurate peak position. Notably, the peak position of *SN₁* mode shows a linear relationship with both the temperature and laser power overall (Fig. 3h, k). By fitting the temperature-dependent Raman shift with Eq. (1), the first-order temperature coefficient χ of *SN₁* mode was determined to be −0.01169 ± 0.00069 cm⁻¹/K (Fig. 3h). On the basis of the relationship between the *SN₁* mode frequency shift and incident laser power (Fig. 3k), the slope δω/δP was extracted to be −0.55471 ± 0.01978 cm⁻¹/mW by linear fitting, which is defined as the laser power-dependent coefficient of *SN₁* mode.

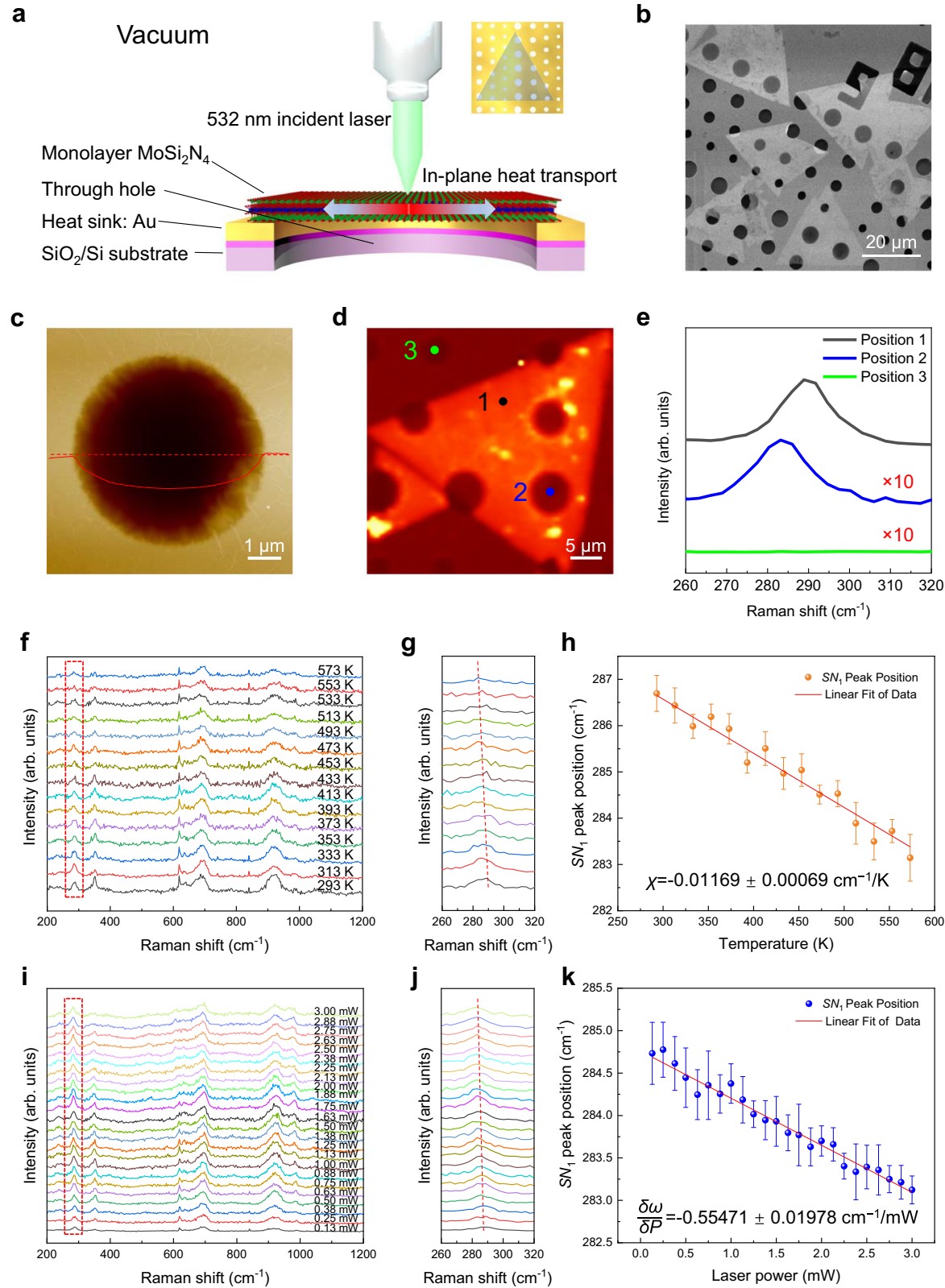

According to the Fourier heat transfer equation, the heat conduction across monolayer $MoSi_2N_4$ can be expressed by[31,39]

$$\frac{\partial Q}{\partial t} = -\kappa \oint \nabla T \cdot dS \qquad (2)$$

where $Q$ is the amount of heat transferred by monolayer $MoSi_2N_4$ during the time $t$ through the surface with cross-section area $S$, $T$ is the absolute temperature, and $\kappa$ is the thermal conductivity. Considering that the radial heat flow propagates from the center of the suspended $MoSi_2N_4$ region toward the hole edge, the thermal conductivity $\kappa$ of

**Fig. 3 | Thermal conductivity measurements of the suspended monolayer MoSi$_2$N$_4$ crystals by optothermal Raman technique. a** Schematic of the optothermal Raman measurements showing suspended MoSi$_2$N$_4$ crystals on Au-deposited holey substrate under 532 nm excitation laser. Inset: macroscopic schematic of the sample on the substrate. **b** Scanning electron microscopy (SEM) image of monolayer MoSi$_2$N$_4$ crystals on Au-deposited SiO$_2$/Si substrate with an array of holes -3, 4, 5, 6 μm in diameter. **c** AFM image of a suspended monolayer MoSi$_2$N$_4$. **d** Raman mapping of the intensity of $SN_1$ mode. Positions 1, 2 and 3 represent the monolayer MoSi$_2$N$_4$ supported on the substrate, the monolayer MoSi$_2$N$_4$ suspended over the hole and the through hole without MoSi$_2$N$_4$, respectively. **e** $SN_1$ mode Raman spectra at different positions in (**d**). **f–h**, Raman spectra (**f**) and $SN_1$ mode (**g**) evolutions with temperature for a suspended monolayer MoSi$_2$N$_4$ crystal and the corresponding temperature-dependent peak position of $SN_1$ mode with a linear fit (**h**). **i–k** Raman spectra (**i**) and $SN_1$ mode (**j**) evolutions with the laser power for a suspended monolayer MoSi$_2$N$_4$ crystal and the corresponding laser power-dependent peak position of $SN_1$ mode with a linear fit (**k**). The red dotted lines in **g** and **j** represent the shift trends of $SN_1$ mode. Error bars in **h** and **k** represent standard deviations from Voigt function fitting in identifying the accurate Raman peak positions.

monolayer MoSi$_2$N$_4$ at room temperature can be evaluated by[14,31,39]

$$\kappa = \alpha \cdot \chi \left(\frac{1}{2\pi h}\right)\left(\frac{\delta\omega}{\delta P}\right)^{-1} \qquad (3)$$

where $\alpha$ (5.52%) is the optical absorption coefficient of 532 nm laser by the suspended monolayer MoSi$_2$N$_4$, $\chi$ is the first-order temperature coefficient extracted from Fig. 3h, $h$ is the thickness of monolayer MoSi$_2$N$_4$ (1.07 nm), $\omega$ is the $SN_1$ mode frequency, $P$ is the external laser power, and $\delta\omega/\delta P$ is the laser power-dependent Raman shift coefficient extracted from Fig. 3k. This yields a $\chi/(\delta\omega/\delta P)$ of 0.021074 ± 0.000492 mW·K$^{-1}$, and consequently a $\kappa$ of 173.03 ± 4.04 W·m$^{-1}$·K$^{-1}$ (see Supplementary Note 2 for details of statistical errors evaluation), which agrees well with the theoretical values, ranging from ~224 to 439 W·m$^{-1}$·K$^{-1}$ [27,29,30] depending on the calculation methods. Notably, this measured value is much higher than those of the 2D semiconductors reported so far including MoS$_2$, WS$_2$, MoSe$_2$, WSe$_2$, Bi$_2$O$_2$Se and black phosphorus, typically less than 100 W·m$^{-1}$·K$^{-1}$ (Fig. 4a and Supplementary Table 1). We also performed optothermal Raman measurements on the same sample in nitrogen gas environment (Supplementary Fig. 9). The extracted thermal conductivity is 227.05 ± 4.32 W·m$^{-1}$·K$^{-1}$, which is ~54 W·m$^{-1}$·K$^{-1}$ higher than the value in vacuum, suggesting that heat convection could not be neglected during the measurements.

We also performed thermal conductivity measurements on suspended monolayer MoSi$_2$N$_4$ with varying hole diameters (3, 4, 5, 6 μm). As shown in Supplementary Fig. 10 and Supplementary Table 2, the measured thermal conductivity increases significantly with increasing the diameter of hole because the low frequency phonons with larger mean free path are excited[17] and it tends to converge under 6-μm-diameter hole. This further confirms that 6-μm-diameter hole is suitable for the accurate measurements of the thermal conductivity of monolayer MoSi$_2$N$_4$. In addition, taking into account the variation of samples in terms of quality, we measured 10 monolayer MoSi$_2$N$_4$ samples from different batches in vacuum. The extracted average thermal conductivity is 171.32 ± 10.86 W·m$^{-1}$·K$^{-1}$ (Supplementary Fig. 11 and Supplementary Table 3).

**Mechanism of unusually high thermal conductivity of MoSi$_2$N$_4$**

According to Slack's criteria, the HTC crystals generally have the following characteristics: low average atomic mass, simple crystal structure (small number of atoms per primitive crystallographic unit cell, $N$), strong interatomic bonding (high Young's modulus) and low anharmonicity. The phonon dispersion curves of crystals with $N=1$ have only acoustic branches, while for $N \geq 2$ there will always be three or more optical branches, which are usually believed to have very small group velocities and hence contribute little to thermal conductivity. Thus, to find HTC crystals, Slack suggested that the candidates should be restricted to structure with small $N$[8].

We compared the thermal conductivities of monolayer MoSi$_2$N$_4$ and other 2D semiconductors from the above first three characteristics. As shown in Fig. 4b and Supplementary Table 1, the thermal conductivities of reported 2D semiconductors decrease with increasing $N$ from 3 to 8 in general. In sharp contrast, the thermal conductivity

of monolayer MoSi$_2$N$_4$ ($N=7$) is unusually high, which deviates from the Slack's criteria. Moreover, monolayer MoSi$_2$N$_4$ also shows much higher thermal conductivity than those with the similar average atomic mass ($\bar{M}$) (Fig. 4c). Young's modulus is a macroscopic expression of the strength of chemical bonds of materials[48,49]. It's interesting to note that the 2D semiconductor with a higher Young's modulus usually has a higher thermal conductivity (Fig. 4d and Supplementary Table 4), suggesting that the strong interatomic bonding plays a key role in the high thermal conductivity of 2D materials including monolayer MoSi$_2$N$_4$.

To date, the theoretical thermal conductivity of monolayer MoSi$_2$N$_4$ has been calculated mainly by two methods. One is a new method based on machine learning interatomic potentials (MLIPs)[27], which can deal with complex crystal structures with large number of atoms to reveal phonon transport with high efficiency and accuracy, and the other is a widely used first-principles calculation based on density functional theory (DFT)[29,30,50]. In order to elucidate the mechanism of high thermal conductivity of monolayer MoSi$_2$N$_4$, we studied its phonon transport properties by solving the phonon Boltzmann transport equation (BTE) based on first-principles calculations. The crystal structure of monolayer MoSi$_2$N$_4$ was fully optimized by the Vienna ab initio simulation package (VASP) based on DFT. Note that the N atoms have two different atomic occupancy sites, denoted as N_1 (inner layer) and N_2 (outer layer) (Fig. 5a). As shown in Fig. 5b, there are twenty-one phonon branches, including three acoustic branches and eighteen optical branches. Supplementary Fig. 12 shows the atomic projected phonon dispersions of monolayer MoSi$_2$N$_4$. The corresponding partial phonon density of states (PDOS) indicates that vibrational frequencies of Mo atoms are mostly below 15 THz due to its greater atomic mass, while Si and N atoms contribute to the whole frequency range (Fig. 5c). We further calculated the Debye temperature $\theta$ (436.45 K) and frequency-dependent Grüneisen parameter $\gamma$ (0.77 at 300 K) (Fig. 5d). Then, the relationship between the thermal conductivity $\kappa$ and intrinsic parameters of monolayer MoSi$_2$N$_4$ was discussed according to the formula[8]

$$\kappa = \frac{B\bar{M}\delta\theta^3}{T\gamma^2} \qquad (4)$$

where $B$ is some constant, $T$ is the absolute temperature (here we focus on room temperature, 300 K), $\bar{M}$ is average atomic mass, and $\delta^3$ is the average volume occupied by one atom of the crystal. For comparison, the parameters of the reported 2D semiconductors were listed in Supplementary Table 5. We plotted thermal conductivities with $\bar{M}\delta\theta^3/\gamma^2$ as the scaling parameter for various 2D semiconductors (Fig. 5e). Note that the thermal conductivity is positively correlated with $\bar{M}\delta\theta^3/\gamma^2$ for all the investigated materials, among which MoSi$_2$N$_4$ has the highest scaling parameter and possesses the highest thermal conductivity.

We further analyzed the relationship between the thermal conductivity and scaling parameter $\bar{M}\delta\theta^3/\gamma^2$. The $\delta$ values of these 2D materials range from 2.17 to 2.85 Å, which thus can be considered as a constant approximatively (Supplementary Table 5). Moreover, $\bar{M}$ of MoSi$_2$N$_4$ is only larger than that of Ti$_2$CO$_2$ among all the investigated

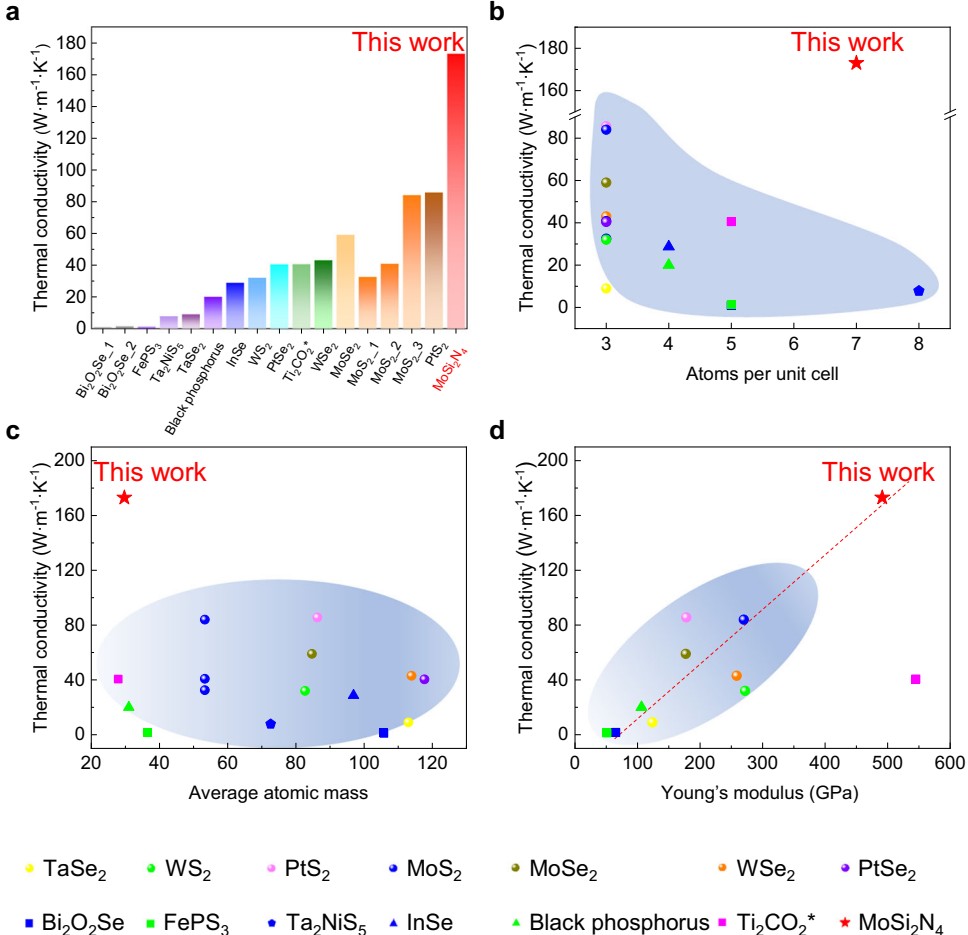

**Fig. 4 | Unusually high thermal conductivity of monolayer MoSi₂N₄ crystals.** **a** Comparison of thermal conductivities of monolayer MoSi₂N₄ and various 2D semiconductors. **b**–**d** Comparisons of the thermal conductivities *versus* atoms per unit cell (**b**), average atomic mass (**c**), and Young's modulus (**d**) for various 2D semiconductors. Note: *The thermal conductivity of Ti₂CO₂ is a theoretical value. For more details, see Supplementary Tables 1 and 4.

2D semiconductors, indicating that a small $\bar{M}$ may not be a key parameter for the high thermal conductivity of MoSi₂N₄. In contrast, Debye temperature $\theta$ and Grüneisen parameter $\gamma$ both show big variations for different materials, which are in the range of 16–500 K and 0.77–2.34, respectively. Importantly, MoSi₂N₄ has the second largest $\theta$ of 436.45 K and the smallest $\gamma$ of 0.77 among them, which are in favor of a high thermal conductivity in terms of Eq. (4). Thus, the large value of $\theta^3/\gamma^2$ plays the most important role for the high thermal conductivity of MoSi₂N₄. In general, a small Grüneisen parameter indicates the small anharmonicity, which usually results from strong chemical bonding[51,52]. A high Debye temperature $\theta$ usually stems from low atomic mass and strong chemical bonding, which will lead to lower phonon-phonon scattering rates as fewer phonon modes are active at a given temperature[8,53,54]. These suggest that the high thermal conductivity of MoSi₂N₄ is strongly dependent on its strong chemical bonding, i.e., high Young's modulus, consistent with the results in Fig. 4d.

As reported in our previous work[20], the outmost Si-N bonds are much stronger than the inner Mo-N bonds and they are mainly responsible for the high Young's modulus of MoSi₂N₄. Thus, we further analyzed the contribution of the outmost Si-N bilayers to the phonon dispersions and PDOS of monolayer MoSi₂N₄. In addition to the three acoustic branches, the two optical branches with the lowest frequency also contribute to the lattice thermal conductivity of MoSi₂N₄ due to the large slopes in the long-wavelength limit (Fig. 5b, c), corresponding to the large phonon group velocities (more than 10 km·s⁻¹ below ~15 THz, Supplementary Fig. 13). It is worth noting that the group

velocities of monolayer MoSi₂N₄ are larger than those of monolayer MoS₂ (6.6 km·s⁻¹)[55], which is one of the main reasons for its high lattice thermal conductivity. We further extracted the atomic contributions for these two optical branches Mo (0%), Si (66.2%), N_1 (1.2%) and N_2 (32.7%) in the long-wavelength limit. Note that the contributions of these two branches to the lattice thermal conductivity of MoSi₂N₄ dominantly stem from the outmost Si-N bilayers. We also calculated the thermal conductivities of monolayer MoSi₂N₄ and InSe-type Si₂N₂ at 100-600 K (Fig. 5f). It can be found that the thermal conductivities of MoSi₂N₄ and Si₂N₂ at 300 K are 492.7 W·m⁻¹·K⁻¹ and 819.5 W·m⁻¹·K⁻¹, respectively, and show the same trend with the temperature rise. Therefore, the outmost Si-N bilayers play a dominant role in the excellent thermal conduction of monolayer MoSi₂N₄, which overwhelm the detrimental influence of complex crystal structure.

## Discussion

The finding of unusually high thermal conductivity in monolayer MoSi₂N₄ not only establishes this material with simultaneously high charge carrier mobility as a benchmark 2D semiconductor for next-generation electronic and optoelectronic devices but also provides an insight into the design of 2D materials for efficient heat conduction. However, the measured thermal conductivity of monolayer MoSi₂N₄ in this work is lower than the theoretical calculation result, which can be ascribed to the following two facts. First, the MoSi₂N₄ crystal has a certain concentration of thermodynamic equilibrium point defects such as N vacancies observed by iDPC-STEM imaging (Supplementary Fig. 14). Similar to the defects in other 2D materials[56,57], the thermal

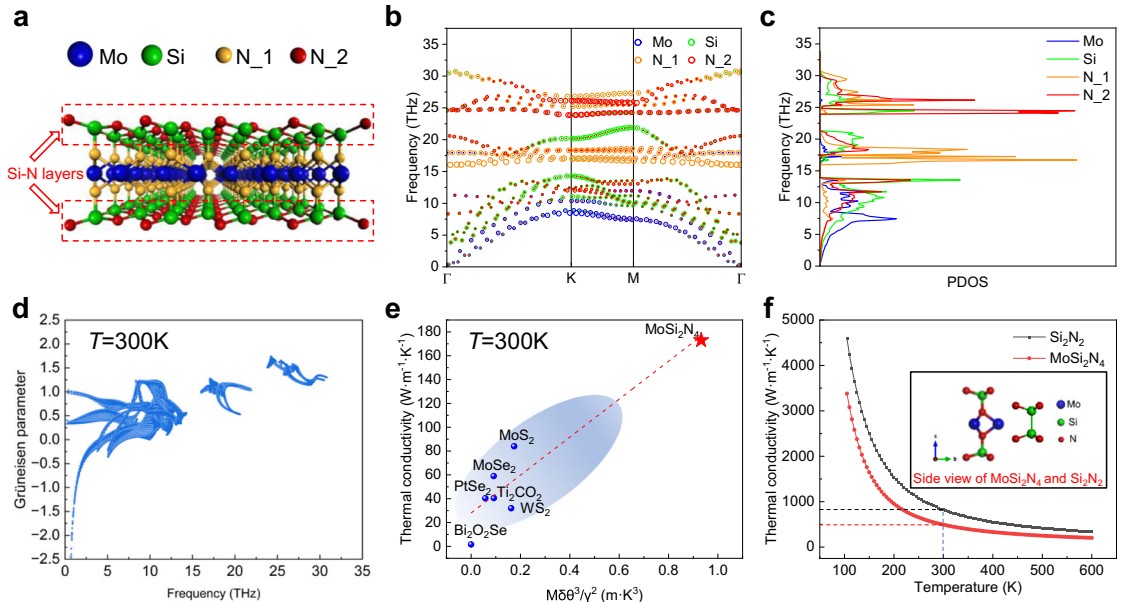

**Fig. 5 | Theoretical analyses on the unusually high thermal conductivity of monolayer MoSi$_2$N$_4$. a** Schematic of atomic model of monolayer MoSi$_2$N$_4$, where N_1 and N_2 represent two different nitrogen atomic occupancies, respectively. **b, c** Calculated phonon dispersions (**b**) and PDOS (**c**) of Mo, Si, N_1, and N_2 in monolayer MoSi$_2$N$_4$, respectively. **d** Frequency-dependent Grüneisen parameter of monolayer MoSi$_2$N$_4$ at 300 K. **e** The thermal conductivities at 300 K versus the scaling parameter $\bar{M}\delta\theta^3/\gamma^2$ for various 2D semiconductors. **f** Calculated in-plane lattice thermal conductivities of monolayer MoSi$_2$N$_4$ (red line) and monolayer Si$_2$N$_2$ (black line) from 100 to 600 K. Inset: side view of the atomic structure model of monolayer MoSi$_2$N$_4$ and Si$_2$N$_2$.

conduction of Si-N bilayers degrades significantly with increasing the density of N vacancies (Supplementary Fig. 15 and Supplementary Note 3). Moreover, N_2 vacancy plays a major role in the high-frequency phonon scattering, while the N_1 vacancy has relatively little influence. Second, the wrinkles in MoSi$_2$N$_4$ (Supplementary Fig. 16), as a form of out-of-plane torsional deformation, can result in strong phonon localizations and enhanced phonon scattering, which consequently reduce the thermal conductivity in analogy with that in other 2D materials[58,59]. Fabricating wafer-scale single crystals of monolayer MoSi$_2$N$_4$ with high quality is the key to utilize its high thermal conductivity and charge carrier mobility for next-generation electronic and optoelectronic devices in the future.

## Methods
### CVD growth and transfer of monolayer MoSi$_2$N$_4$ crystals
Monolayer MoSi$_2$N$_4$ crystals were grown by CVD at 1065 °C with Cu/Mo bilayer foils as the substrate, NH$_3$ gas as the nitrogen source, and quartz as the silicon source. In ultrahigh-purity hydrogen (H$_2$) atmosphere (200 sccm, 99.999% purity), the Cu/Mo stacked substrate was first heated to 1090 °C in 50 minutes to make Cu thoroughly melt and then the temperature was dropped to 1065 °C rapidly, which enabled a good interface between Cu layer and Mo layer. After that, NH$_3$ (3 sccm, 99.0% purity) was introduced and triangular MoSi$_2$N$_4$ crystals with an average size of more than 20 μm were obtained after 1.5 hours growth. Besides, more than 3 hours were required to obtain MoSi$_2$N$_4$ films. In order to transfer MoSi$_2$N$_4$ crystals/films to target substrates, poly (methyl methacrylate) (PMMA, 4 wt% in ethyl lactate) was spin-coated (3000 rpm, 1 minute) onto the surface of MoSi$_2$N$_4$ followed by curing on a hot plate at 120 °C for 10 minutes. After Cu layer was etched away by 0.15 M (NH$_4$)$_2$S$_2$O$_8$ solution, PMMA/MoSi$_2$N$_4$ stack was separated from the growth substrate and scooped up with target substrate, followed by air-drying at 60 °C for 10 minutes and then baking at 120 °C for 10 minutes. Finally, PMMA was dissolved in acetone, leaving MoSi$_2$N$_4$ on the through-hole Au/SiO$_2$/Si substrate or other target substrates (e. g. SiO$_2$/Si, quartz, TEM grids, etc.) for the subsequent characterizations and measurements.

### Structure and optical properties characterizations
The morphology of monolayer MoSi$_2$N$_4$ samples was characterized by optical microscope (Nikon LV100D), AFM (MultiMode 8, Bruker, Inc.), and SEM (Verios G4 UC, 0.6 nm@15 kV). The in-plane and cross-plane atomic level structures were characterized by TEM (FEI Titan Cubed Themis G2 300). The SAED patterns were obtained on a TEM operating at 120 kV (FEI Tecnai T12). Optical transmittance spectrum was measured in a Varain Cary 5000 spectrometer.

### Thermal conductivity measurements
The through-hole substrates for thermal conductivity measurements were fabricated by Suzhou YW Mems. Periodic through-hole-arrays of 3, 4, 5, 6 μm in diameter were patterned on Si substrate with a 285-nm-thick SiO$_2$ layer, and 100-nm-thick Au film was evaporated as a heat sink on the through-hole substrate prior to MoSi$_2$N$_4$ transfer. The optothermal micro-Raman spectroscopy measurements were performed on WITec alpha300R confocal Raman microscopy (Oxford WITec) with in-situ thermal stages of KT-Z165M4LT (room temperature to 400 °C in vacuum, Zhengzhou Ketan Instrument Equipment Co., LTD.) and LINKAM HFS600E-PB4 (-196 °C to 600 °C in nitrogen gas environment, Linkam Scientific Instruments LTD.). A long-working-distance 50 × objective lens with a numerical aperture (NA) of 0.55 was used in the measurements. A 532 nm diode-pumped solid-state laser (cobalt laser) was used as excitation source with a laser spot size ~1.18 μm ($D$ = 1.22 $\lambda$/NA). The signals were detected using a charge-coupled-device (CCD) thermoelectrically cooled to -60 °C. During the measurements of temperature-dependent Raman spectra, the laser power was set at 200 μW to avoid local laser heating effect and structural damages. More than 5 Raman spectra were collected to ensure the credibility and repeatability of the results for each sample.

### Theoretical calculations of phonon dispersions and thermal conductivity
First-principles calculations were performed by using the Vienna ab initio simulation package (VASP)[60-62] based on DFT[63]. The projected augmented wave (PAW) method[64] and generalized gradient

approximation with the Perdew-Burke-Ernzerh exchange-correlation functional (GGA-PBE)[61,65] were used. The plane-wave cutoff energy of 500 eV was used with a $15 \times 15 \times 1$ $k$-mesh. A large vacuum space was set to 20 Å to avoid the interactions between the neighboring layers. Both the lattice constants and internal atomic positions of monolayer $MoSi_2N_4$ and $Si_2N_2$ were allowed to relax and the convergence criteria for energies and forces were $10^{-8}$ eV and $10^{-8}$ eV/Å, respectively. In order to understand the transport properties of phonons, phonon dispersions and the second-order harmonic (2nd) interatomic force constants (IFCs) were calculated via the finite displacement method within the Phonopy package[66]. The third-order anharmonic (3rd) IFCs with a certain cutoff distance were calculated within the thirdorder package[67]. A $4 \times 4 \times 1$ supercell (112 atoms) with a $3 \times 3 \times 1$ $k$-mesh was used to calculate the second- and third-order IFCs. The lattice thermal conductivity and phonon properties of monolayer $MoSi_2N_4$ and $Si_2N_2$ were calculated by solving the phonon Boltzmann transport equation, as implemented in the ShengBTE package[67]. Different q-grids and cutoff distances were used to test the convergence of the thermal conductivities of $MoSi_2N_4$ and $Si_2N_2$ (Supplementary Fig. 17). According to the convergence tests, we used a q-grid of $111 \times 111 \times 1$ and a cutoff distance of 8 atoms for $MoSi_2N_4$, and a q-grid of $121 \times 121 \times 1$ and a cutoff distance of 9 atoms for $Si_2N_2$. The Born effective charges and dielectric constants were considered during first-principles calculations.

### Reporting summary
Further information on research design is available in the Nature Portfolio Reporting Summary linked to this article.

## Data availability
The authors declare that all the data supporting the results of this study can be found in the paper and its Supplementary Information file. The detailed data for the study is available from the corresponding author upon request.

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

## Acknowledgements

We thank Mr. J. Lai for advice on first-principles calculations of lattice thermal conductivity and Mr. R. Liu for the help with constructing the atomic structure model of $MoSi_2N_4$. We also acknowledge Prof. C. Liu and Prof. L. Zhang for providing Raman measurement system and in-situ vacuum thermal stage. This work was financially supported by the National Natural Science Foundation of China (Nos. 52188101, 52122202), the Key Research Program of Frontier Sciences of the Chinese Academy of Sciences (No. ZDBS-LY-JSC027), the LiaoNing Revitalization Talents Program (No. XLYC2201003), and the Institute of Metal Research, Chinese Academy of Sciences (Project Young Merit Scholars, No. 2019000178).

## Author contributions

W.R. conceived and supervised the project; W.R., C.X., and C.H. designed the experiments; C.H. performed growth experiments, AFM, Raman, UV–vis optical property, and thermal conductivity measurements; C.C. performed the theoretical calculations. J.T. and Z.L. performed TEM characterizations; T.Z. and S.S. helped with optical property measurements; C.H., C.X., C.C., and W.R. analyzed the data and wrote the manuscript with input from all authors; H.-M.C. provided advice on the project.

## Competing interests

The authors declare no competing interests.
