## [Peer Review File · Nature Communications]

Unusually high thermal conductivity in suspended monolayer MoSi₂N₄REVIEWER COMMENTS

Reviewer #1 (Remarks to the Author):

In this work, for the first time authors measured the thermal conductivity of the monolayer MoSi₂N₄ grown by chemical vapor deposition (CVD) using a noncontact optothermal Raman technique. Their first-principles within the PBE/GGA calculations reveal that such unusually high thermal conductivity is due to high Debye temperature and small Grüneisen parameter of MoSi₂N₄. The studied 2D film has been very recently experimentally fabricated, and is currently among the most interesting 2D materials, and such that this study is timely and the obtained results can be potentially useful and attractive. The acquired results propose that the MoSi₂N₄ nanosheet exhibits many advantages which could be a very promising material for nanoelectronics and thermal management. The article covers an important field of research, the exploration of new 2D materials beyond the graphene, as such this manuscript is an important contribution and I can recommend this manuscript for publication, after the authors address the following concerns:

- 1- Please also highlight that the high thermal conductivity also originates from the remarkably high elastic modulus and phonon group velocities as well.
- 2- Were the Born effective charges and dielectric constants considered in the theoretical solution?
- 3- For the evaluation of anharmonic force constants, what was the cutoff distance? Has it been tested for convergence?
- 4- How were the statistical errors evaluated in the experimental tests? What are the sources? How many samples have been tested? more details should be given.
- 5- Please also comment on the effects of substrate on the measured thermal conductivity. In the title, please also clearly mention whether the samples were suspended or supported.
- 6- Authors should comment on an alternative solution based on MLIPs as presented in Nano Energy 82,105716, 2021 which are of DFT accuracy but MD efficiency and allow bridging first principles based models up to continuum models.

Reviewer #2 (Remarks to the Author):

Monolayer MoSi₂N₄ is an emerging 2D material with versatile properties and perspective applications predicted by theoretical calculations. Many excellent properties require experimental verification. In this manuscript, the authors investigated and revealed the unusually high thermal conductivity of monolayer MoSi₂N₄ by using a noncontact optothermal Raman technique and first-principles calculations. This study is comprehensive and instructive, with a high expectation of readership. I recommend publishing the manuscript after revisions. Several factors are provided as below that should be addressed and corrected to enhance the quality and readability.

- 1) The authors explored various potential factors affecting thermal conductivity measurements, such as the hole size of the substrate, temperature, and laser power. However, there is no discussion regarding the laser spot size, which has the potential to impact both spatial resolution and temperature distribution. It is necessary to explicitly explain the choice of laser spot size (1.18 μm in this manuscript).
- 2) In this manuscript, the difference between the measured thermal conductivity and theoretical calculation result is attributed, in part, to the presence of a certain concentration of N vacancies. One would expect whether any the impact of the position and concentration

of vacancy on the thermal conductivity.

3) The theoretical thermal conductivity is calculated by solving the phonon Boltzmann transport equation, as implemented in the ShengBTE package. The results for the convergence of the thermal conductivity with respect to the q-grid has to be presented.

4) There is an incorrect description of “The third-order harmonic (3rd) IFCs ...” in Line 361-363. Harmonic vibration is only referred to second-order interatomic force constants. The term “harmonic” should be corrected to “anharmonic”.

5) Besides, in the 3rd IFCs calculations, the authors only considered the interactions up to 5 Å, is this cutoff distance accurate enough?

Reviewer #3 (Remarks to the Author):

The manuscript written by Chengjian He et al. had measured thermal conductivity of $\sim 173 \text{ W}\cdot\text{m}^{-1}\cdot\text{K}^{-1}$ at room temperature for monolayer MoSi₂N₄ grown by chemical vapor deposition. Noncontact optothermal Raman technique and First-principles calculations reveal that such unusually high thermal conductivity benefits from the high Debye temperature and small Grüneisen parameter of MoSi₂N₄. They are strongly dependent on the high Young's modulus induced by the outmost Si-N layers, which is interesting. However, there are some questions need to be answered.

1. Among various vdW layered 2D semiconductor, why do the authors choose MoSi₂N₄? Does it have any specific superior performance parameters that have been experimentally verified? Or some potentially huge application? The author doesn't seem to explain it very well.
2. “the known 2D semiconductors” in Abstract is a very large range, and the use of words like “highest” is discouraged unless there is solid evidence and a complete comparison. Kindly remove such words.
3. The references are only 16/65 in the past three years, and some references are too old. Proposed update.
4. Voigt function fitting and Linear fitting inevitably have errors, so the final thermal conductivity value of monolayer MoSi₂N₄ is a certain range, the more scientific form should be $A \pm B \text{ W}\cdot\text{m}^{-1}\cdot\text{K}^{-1}$.
5. As mentioned by the authors, the diameter of the hole also affects the heat dissipation of MoSi₂N₄. How to determine if a hole with a diameter of 6 μm is large enough for heat flow transfer from the center to the edge of monolayer MoSi₂N₄? And the substrate is a periodic through-hole arrays of 3, 4, 5, 6 μm in diameter. Is there any experimental data with smaller holes to support the author's judgment?
6. Is there a more detailed and physical explanation for the Raman peak redshift of MoSi₂N₄ suspended and supported on Au/SiO₂/Si substrates in Figure 3e?
7. Why might point defects reduce the thermal conduction of Si-N layers? Are other material preparation methods, such as mechanical stripping, expected to improve the situation?
8. Why the wrinkles in MoSi₂N₄ can enhance phonon scattering? And is “enhance phonon scattering” necessarily related to “reduce the thermal conductivity”?
9. It is suggested the authors to check their manuscript carefully and thoroughly to avoid some minor typical mistakes and mistypes. Such as in Figure 3k, what is “laser powder-dependent”?
10. In Figure 3k, what's the difference if we change the wavelength of the excitation laser?

Response to reviewers' comments

Reviewer #1

In this work, for the first time authors measured the thermal conductivity of the monolayer MoSi_2N_4 grown by chemical vapor deposition (CVD) using a noncontact optothermal Raman technique. Their first-principles within the PBE/GGA calculations reveal that such unusually high thermal conductivity is due to high Debye temperature and small Grüneisen parameter of MoSi_2N_4 . The studied 2D film has been very recently experimentally fabricated, and is currently among the most interesting 2D materials, and such that this study is timely and the obtained results can be potentially useful and attractive. The acquired results propose that the MoSi_2N_4 nanosheet exhibits many advantages which could be a very promising material for nanoelectronics and thermal management. The article covers an important field of research, the exploration of new 2D materials beyond the graphene, as such this manuscript is an important contribution and I can recommend this manuscript for publication, after the authors address the following concerns:

Response: We thank the reviewer very much for the positive comments and constructive suggestions, which have helped us greatly improve the quality of our manuscript.

1. Please also highlight that the high thermal conductivity also originates from the remarkably high elastic modulus and phonon group velocities as well.

Response: We thank the reviewer very much for the constructive suggestion.

We fully agree with the reviewer that the high thermal conductivity also originates from the remarkably high elastic modulus and phonon group velocities. The importance of high elastic modulus has been highlighted in our original manuscript. Actually, we also calculated the phonon group velocities of monolayer MoSi_2N_4 , as shown in Supplementary Fig. 13. According to the reviewer's suggestion, we have highlighted the importance of high phonon group velocities for the high thermal conductivity as well in our revised manuscript.

2. Were the Born effective charges and dielectric constants considered in the theoretical

solution?

Response: We thank the reviewer very much for the insightful comment.

In the shengBTE.py package, the born effective charge and dielectric constants are important parameters for calculating the lattice thermal conductivity and are used to account for the long-range electrostatic interactions, especially for polar compounds. In our calculations, we used VASP to calculate the Born effective charge and dielectric constant of monolayer MoSi₂N₄ and both of them were considered in our theoretical solution, as shown in Fig. R1.

```
#CONTROL for ShengBTE
&allocations
  nelements=3
  natoms=7
  ngrid(:)=111 111 1
&end
&crystal
  lfactor=0.100000
  lattvec(:,1)=2.9102481010437544 -0.0000793793438826 0.0000000000000000
  lattvec(:,2)=-1.4551927950503551 2.5203091020434223 0.0000000000000000
  lattvec(:,3)=0.0000000000000000 0.0000000000000000 27.0002463739445844
  elements= "Mo" "Si" "N"
  types= 1 2 2 3 3 3 3
  positions(:,1)=0.3333432880869795 0.6666567119130188 0.5000000000000000
  positions(:,2)=0.666658044671226 0.333341955328760 0.6109190060388051
  positions(:,3)=0.666658044671226 0.333341955328760 0.3890809939611973
  positions(:,4)=0.6666744895991493 0.333255104008502 0.4538064063640140
  positions(:,5)=0.6666744895991493 0.333255104008502 0.5461935936359881
  positions(:,6)=0.9999930618902348 0.0000069381097661 0.3703234310123362
  positions(:,7)=0.9999930618902348 0.0000069381097661 0.6296765689876629
  epsilon(:,1)=3.97433242 -0.00008557 0.00000000
  epsilon(:,2)=-0.00008557 3.97423361 0.00000000
  epsilon(:,3)=0.00000000 0.00000000 1.40992944
  born(:,1,1)=1.03163167 0.00002490 0.00000000
  born(:,2,1)=0.00002490 1.03166042 0.00000000
  born(:,3,1)=0.00000000 0.00000000 -0.12897129
  born(:,1,2)=3.28930793 -0.00013392 0.00023634
  born(:,2,2)=-0.00013392 3.28915330 -0.00013645
  born(:,3,2)=0.00023995 -0.00013854 0.74232502
  born(:,1,3)=-0.93377536 -0.0000762 0.00016879
  born(:,2,3)=-0.0000762 -0.93378416 -0.00009745
  born(:,3,3)=0.00015958 -0.00009214 -0.59928526
  born(:,1,4)=-2.87134841 0.00012909 0.00003204
  born(:,2,4)=0.00012909 -2.87119935 -0.00001850
  born(:,3,4)=0.00171378 -0.00098945 -0.07855412
  scell(:)=4 4 1
```

Figure. R1 | Printscreen of Born effective charges and dielectric constants considered in the theoretical calculations.

3. For the evaluation of anharmonic force constants, what was the cutoff distance? Has it been tested for convergence?

Response: We thank the reviewer very much for the insightful comments.

Regarding to the calculation of the thermal conductivity of MoSi₂N₄ in our work, we used the cutoff distance of 5 atoms and the q-grid of 81 × 81 × 1 as the convergence standard, which are moderate values compared with those used in the reported work. The calculated thermal conductivity was 450.3 W·m⁻¹·K⁻¹ at 300 K, which is close to

the reported values [*Nano Energy* **82**, 105716 (2021); *ACS Appl. Mater. Interfaces* **13**, 45907 (2021)].

According to the review's suggestions, we further tested the convergence of the thermal conductivities of MoSi_2N_4 and Si_2N_2 by using different q-grid and cutoff distance. As shown in Fig. R2, the thermal conductivity of MoSi_2N_4 and Si_2N_2 converges to $492.7 \text{ W}\cdot\text{m}^{-1}\cdot\text{K}^{-1}$ and $819.5 \text{ W}\cdot\text{m}^{-1}\cdot\text{K}^{-1}$ at 300 K, respectively, when 8-atoms cutoff distance and $111 \times 111 \times 1$ q-grid were used for MoSi_2N_4 and 9-atoms cutoff distance and $121 \times 121 \times 1$ q-grid were used for Si_2N_2 . This indicates that the intrinsic thermal conductivity was underestimated slightly for MoSi_2N_4 in our original calculation. The ultrahigh thermal conductivity of Si_2N_2 suggests that the outer Si-N layer in MoSi_2N_4 plays an important role for thermal conduction.

We have updated the related data and added Fig. R2 in our revised manuscript.

Figure R2 | The convergence tests of cutoff distance and q-grid of MoSi_2N_4 and Si_2N_2 at 300 K. a,b, The convergence tests of cutoff distance of MoSi_2N_4 with a q-grid

of $111 \times 111 \times 1$ (a) and Si_2N_2 with a q-grid of $121 \times 121 \times 1$ (b). c,d, The convergence tests of q-grid of MoSi_2N_4 with cutoff distances of 5, 8, 10 atoms (c) and Si_2N_2 with cutoff distances of 5, 9, 10 atoms (d).

4. How were the statistical errors evaluated in the experimental tests? What are the sources? How many samples have been tested? more details should be given.

Response: We thank the reviewer very much for the kind comments and valuable suggestions.

There are two kinds of sources for the statistical errors in our experimental tests, including the error of Voigt fitting in precise identification of the Raman peak position and the error of linear fitting of extracting first-order temperature and power coefficients. In our statistical calculation model, we reconsidered these two kinds of errors for evaluating the thermal conductivity.

According to the Error Propagation Formula in Statistics, as $\varepsilon(x_1)$ and $\varepsilon(x_2)$ are the errors of x_1 and x_2 , the statistical error of x_1/x_2 should be:

$$\varepsilon(x_1/x_2) \approx \frac{x_1 \cdot \varepsilon(x_2) - x_2 \cdot \varepsilon(x_1)}{x_2^2} \quad (x_2 \neq 0) \quad \text{Eq. (R1)}$$

Taking the extracted data in our manuscript for example, the first-order temperature coefficient χ and laser power dependent coefficient $\delta\omega/\delta P$ are $-0.01169 \pm 0.00069 \text{ cm}^{-1}/\text{K}$ and $-0.55471 \pm 0.01978 \text{ cm}^{-1}/\text{mW}$, respectively, thus $\chi / \left(\frac{\delta\omega}{\delta P} \right)$ should be $0.021074 \pm 0.000492 \text{ mW/K}$. By substituting this value into Eq. (3) in the manuscript, the thermal conductivity of monolayer MoSi_2N_4 was calculated as $173.03 \pm 4.04 \text{ W} \cdot \text{m}^{-1} \cdot \text{K}^{-1}$.

To obtain more solid statistical results for the thermal conductivity of monolayer MoSi_2N_4 , we measured ten samples of MoSi_2N_4 single crystals from different batches and extracted the average value of thermal conductivities. The same method has been used to evaluate the thermal conductivity of monolayer graphene [*Nat. Nanotech.* **17**, 1258–1264 (2022)]. The obtained thermal conductivity values and the corresponding $\chi / \left(\frac{\delta\omega}{\delta P} \right)$ of the ten MoSi_2N_4 samples are shown in Table R1 and Fig. R3, respectively. The average thermal conductivity of ten samples is $\sim 171.32 \pm 10.86 \text{ W} \cdot \text{m}^{-1} \cdot \text{K}^{-1}$. In our manuscript, we provided a representative value of $173.03 \pm 4.04 \text{ W} \cdot \text{m}^{-1} \cdot \text{K}^{-1}$ as the measured thermal conductivity of suspended monolayer MoSi_2N_4 .

We have added the detailed description of errors calculation process, Figure R3 and Table R1 as Supplementary Note 2, Supplementary Fig. 11 and Supplementary Table 3, respectively, in our revised manuscript.

Figure R3 | The measured thermal conductivities of 10 suspended monolayer MoSi₂N₄ samples from different batches.

Table R1 | The $\chi / (\delta\omega/\delta p)$ and the corresponding thermal conductivity values of 10 suspended monolayer MoSi₂N₄ samples.

Sample number	$\chi / (\delta\omega/\delta p)$ (mW·K ⁻¹)	Thermal conductivity (W·m ⁻¹ ·K ⁻¹)
1	0.021048 ± 0.002448	172.81 ± 20.10
2	0.021649 ± 0.000295	177.75 ± 2.42
3	0.016000 ± 0.000524	131.37 ± 4.30
4	0.017109 ± 0.000025	140.47 ± 0.21
5	0.026097 ± 0.000937	214.28 ± 7.69
6	0.021074 ± 0.000492	173.03 ± 4.04
7	0.023785 ± 0.003257	195.29 ± 26.74
8	0.020653 ± 0.000454	169.58 ± 3.73

9	0.020174 ± 0.002525	165.64 ± 20.73
10	0.021077 ± 0.002269	173.05 ± 18.63

5. Please also comment on the effects of substrate on the measured thermal conductivity. In the title, please also clearly mention whether the samples were suspended or supported.

Response: We thank the reviewer very much for the valuable suggestions.

For 2D materials supported on substrate, they suffer from phonon-defects, phonon-impurities and phonon-boundaries scatterings and heat dissipation along the out-of-plane direction [*J. Phys. Chem. C* **125**, 16129-16135 (2021)]. These factors strongly affect the measurements, leading to an inaccurate lattice thermal conductivity. Suspending the sample can eliminate the thermal coupling between 2D material and the substrate as well as the phonon scatterings caused by the substrate. Moreover, suspending in vacuum will force heat to propagate along the in-plane direction, avoiding the extra out-of-plane heat conduction. Therefore, we conducted measurements using suspended MoSi₂N₄ in vacuum for obtaining more precise values of thermal conductivity. In addition, we designed through holes to avoid the interference from laser light that was reflected and scattered from the bottom of the holes. Otherwise, the interference will also bring inaccuracy to the measurement of thermal conductivity.

We have added more discussions about the effects of substrate on the measured thermal conductivity in our revised manuscript. In addition, according to the reviewer's suggestion, we have revised the title as "Unusually high thermal conductivity in suspended monolayer MoSi₂N₄".

6- Authors should comment on an alternative solution based on MLIPs as presented in *Nano Energy* **82**,105716, 2021 which are of DFT accuracy but MD efficiency and allow bridging first principles based models up to continuum models.

Response: We thank the reviewer very much for the kind suggestion.

We fully agree with the reviewer that this pioneering work based on machine learning

interatomic potentials (MLIPs) [*Nano Energy* **82**, 105716 (2021)] provides a powerful method for studying phonon transport with high efficiency and accuracy, which can deal with complex crystal structures with large number of atoms. Multi-scale computational simulation has made great contributions to the design and prediction of material structure and properties, and brought great inspiration to our experimental work. In our revised manuscript, we have highlighted this work as the advanced solution for the calculation of thermal conductivity. However, limited by our knowledge, we used the classical first-principles calculations by ShengBTE packages to calculate the lattice thermal conductivity of monolayer MoSi₂N₄, the accuracy of which has been widely recognized as well [*ACS Appl. Mater. Interfaces* **13**, 45907-45915 (2021); *New J. Phys.* **23**, 033005 (2021); *J. Solid State Chem.* **315** (2022)].

Reviewer #2:

Monolayer MoSi₂N₄ is an emerging 2D material with versatile properties and perspective applications predicted by theoretical calculations. Many excellent properties require experimental verification. In this manuscript, the authors investigated and revealed the unusually high thermal conductivity of monolayer MoSi₂N₄ by using a noncontact optothermal Raman technique and first-principles calculations. This study is comprehensive and instructive, with a high expectation of readership. I recommend publishing the manuscript after revisions. Several factors are provided as below that should be addressed and corrected to enhance the quality and readability.

Response: We thank the reviewer very much for the positive comments and constructive suggestions, which have helped us greatly improve the quality and readability of our manuscript.

1) The authors explored various potential factors affecting thermal conductivity measurements, such as the hole size of the substrate, temperature, and laser power. However, there is no discussion regarding the laser spot size, which has the potential to impact both spatial resolution and temperature distribution. It is necessary to explicitly explain the choose of laser spot size (1.18 μm in this manuscript).

Response: We thank the reviewer very much for the insightful comments and constructive suggestions.

We fully agree with the reviewer that the laser spot size has the potential to impact both spatial resolution and temperature distribution. For the optothermal Raman measurements of the thermal conductivity of 2D materials, the lasers with a spot size of 0.92-1.42 μm (in diameter) have been used in the reported works, and the corresponding hole size of the substrate was 1.2-6 μm (in diameter) [*Phys. Rev. Appl.* **13**, 034059 (2020); *ACS Appl. Mater. Interfaces* **7**, 25923-25929 (2015); *ACS Nano* **8**, 986-993 (2014); *Nature* **597**, 660-665 (2021)]. In our work, 1.18-μm laser spot and 6-μm hole were used, which were enough to obtain high spatial resolution for accurately determining the sample position and acquiring accurate Raman spectra at the center of the sample.

Figure R4 | Simulated spatial temperature distribution of monolayer MoSi_2N_4 suspended on a 6- μm -diametered hole at an excitation power of 3 mW.

To confirm that this laser spot size is suitable for measurements, we simulated the laser heating (schematic shown in Fig. 3a) using a finite element thermal simulation with COMSOL Multiphysics to obtain the temperature distribution profile of suspended monolayer MoSi_2N_4 . In our experiments, we used a maximum incident laser power of 3 mW with a laser spot size of 1.18 μm and hole diameter of 6 μm . The simulation results indicate that even under the maximum power, the temperature of suspended monolayer MoSi_2N_4 near the edge of the 6- μm -diametered hole reached 300.06 K (ambient temperature) as shown in Fig. R4. This ensures that the heat flow could transfer from the center to the edge of the suspended monolayer MoSi_2N_4 , avoiding the influence of substrate on the accuracy of thermal conductivity. Accordingly, the laser with a spot size of 1.18 μm was used in our measurements.

In our revised manuscript, we have added Figure R4 as Supplementary Fig. 8 and the above related discussions. In addition, the simulation method has been added as Supplementary Note 1. The detailed simulation process is shown below.

The Fourier heat diffusion equations were solved with cylindrical coordinates (2D

axisymmetric configuration) and a Gaussian-shaped beam heat source. The temperature increase of a suspended monolayer MoSi₂N₄ under laser excitation is directly related to its thermal conductivity, assuming that the absorbed heat transfers radially through a small cross-sectional area of the flake from the center to the edge.

As reported in previous works [*J. Phys. Chem. C* **125**, 16129-16135 (2021); *ACS Nano* **8**, 986-993 (2014)], the diffusion of heat in the suspended region can be expressed as the following equation:

$$\kappa \frac{1}{r} \frac{d}{dr} \left[r \frac{dT_1(r)}{dr} \right] + q(r) = 0 \quad \text{for } r < R \quad \text{Eq. (R2)}$$

where κ is the in-plane thermal conductivity of suspended monolayer MoSi₂N₄, $T_1(r)$ is the temperature distribution profile inside the hole (radius R), $q(r)$ is the volumetric optical heating and defined as Eq. (R3)

$$q(r) = \frac{I\alpha}{t} \exp\left(-\frac{r^2}{r_0^2}\right) \quad \text{Eq. (R3)}$$

where $I = P/(\pi r_0^2)$ is the peak laser power per unit area at the center of beam spot, α is the optical absorption coefficient of monolayer MoSi₂N₄ at 532 nm, t is the thickness of monolayer MoSi₂N₄ flake, and $r_0 = 0.59 \mu\text{m}$ is the laser beam radius used in our experimental system.

While outside the hole, the heat transports not only along the flake, but also into the Au/SiO₂/Si substrate, so its heat dissipation can be described as the following equation:

$$\kappa' \frac{1}{r} \frac{d}{dr} \left[r \frac{dT_2(r)}{dr} \right] - \frac{G}{t} [(T_2(r) - T_a)] = 0 \quad \text{for } r > R \quad \text{Eq. (R4)}$$

where κ' is the in-plane thermal conductivity of the supported monolayer MoSi₂N₄, $T_2(r)$ is the temperature distribution profile outside the hole, G is the interfacial thermal conductance between monolayer MoSi₂N₄ and Au/SiO₂/Si substrate, and T_a is the ambient temperature (300 K).

Consequently, the spatial distribution profiles of temperature inside the hole [$T_1(r)$] and outside the hole [$T_2(r)$] could be obtained by solving Eqs. (R2) and (R4), and the results are

$$T_1(r) = c_1 - 2c_2 \ln(r) + c_3 Ei \left[1, \left(\frac{r}{r_0} \right)^2 \right] \quad \text{for } r < R \quad \text{Eq. (R5)}$$

$$T_2(r) = c_4 K_0 \left[0, r \sqrt{\frac{G}{\kappa t}} \right] + T_a \quad \text{for } r > R \quad \text{Eq. (R6)}$$

where c_i are constants to be determined by boundary conditions, K_0 is the zero-order modified Bessel function of the second kind, and Ei is the exponential integral.

Taking into account the suitable boundary conditions ($r = R$ and $r \rightarrow \infty$):

$$T_1(R) = T_2(R) \quad \text{Eq. (R7)}$$

$$T_2(r) = T_a \quad (r \rightarrow \infty) \quad \text{Eq. (R8)}$$

the temperature profile for both inside the hole [$T_1(r)$] and outside the hole [$T_2(r)$] under different laser powers can be expressed with three unknown parameters κ , κ' , and G . For κ' , a reasonable assumption $\kappa' = \kappa$ was made, which has been adopted in the reported works [*J. Phys. Chem. C* **125**, 16129-16135 (2021); *ACS Nano* **8**, 986-993 (2014)]. Therefore, in our COMSOL simulation process, we input $173 \text{ W}\cdot\text{m}^{-1}\cdot\text{K}^{-1}$ as the values of κ and κ' . In addition, in the COMSOL simulation process, we need to input the value of out-of-plane thermal conductivity κ_{\perp} .

Next, we changed the assumed conditions for G and κ_{\perp} in the COMSOL simulations, and the obtained temperature curves (Fig. R5) are basically coincident at the same hole size and excitation power. This result indicates that the interface does not have an influence on the temperature distribution, which is consistent with the conclusions reported in previous work [*J. Phys. Chem. C* **125**, 16129-16135 (2021)]. Thus, we chose a representative interface thermal conductivity (G) value of $100 \text{ MW}\cdot\text{m}^{-2}\cdot\text{K}^{-1}$ and a reasonable κ_{\perp} of $0.173 \text{ W}\cdot\text{m}^{-1}\cdot\text{K}^{-1}$. In fact, under the present experimental conditions, κ_{\perp} of monolayer 2D materials cannot be directly measured. Previous work [*Nature* **597**, 660-665 (2021)] has experimentally confirmed that the κ_{\perp} values of multilayer MoS_2 ($\kappa_{\perp}=57 \pm 3 \text{ mW}\cdot\text{m}^{-1}\cdot\text{K}^{-1}$) and WS_2 ($\kappa_{\perp}=41 \pm 3 \text{ mW}\cdot\text{m}^{-1}\cdot\text{K}^{-1}$) are about 1/900 of their respective in-plane thermal conductivity. Therefore, here we assumed that the value of κ_{\perp} is three orders of magnitude smaller than that of κ for monolayer MoSi_2N_4 .

Fig. R5 | Temperature profiles across the monolayer MoSi₂N₄ at different excitation laser power, interface thermal conductance and out-of-plane thermal conductivity under the hole diameter of 6 μm. a, With a constant out-of-plane thermal conductivity (κ_{\perp}) of 0.173 W·m⁻¹·K⁻¹; b, With a constant interface thermal conductance (G) of 100 MW·m⁻²·K⁻¹.

2) In this manuscript, the difference between the measured thermal conductivity and theoretical calculation result is attributed, in part, to the presence of a certain concentration of N vacancies. One would expect whether any the impact of the position and concentration of vacancy on the thermal conductivity.

Response: We thank the reviewer very much for the insightful comment.

Many previous experimental and theoretical work has studied the effects of vacancy density and vacancy type on the thermal conductivity of 2D materials. For example, the thermal conductivity of suspended graphene with different densities of carbon vacancy has been measured and the relationship between the thermal conductivity and vacancy density has been discussed [*Nanoscale* **8**, 14608-14616 (2016)]. The phonon transport of few-layer MoS₂ nanosheets with different densities of vacancy has been studied as well [*Nanoscale* **13**, 11561-11567 (2021)]. These experiments showed similar results that the thermal conductivity of 2D materials decreases significantly with the increase of vacancy density, especially at low vacancy density. Theoretical works based on DFT and nonequilibrium molecular dynamics (NEMD) calculations also demonstrated that the thermal conductivity of MoS₂ decreases significantly due to the existence of vacancies [*Phys. Rev. Mater.* **4** 014004 (2020); *J. Phys. Chem. C* **119**, 16358-16365

(2015)].

In order to reveal the influence of N vacancies with different positions on the thermal conductivity of monolayer MoSi_2N_4 , we investigated the phonon scattering by point defects based on Klemens P. G.'s perturbation theory [*Int. J. Thermophys.* **8**, 737-750 (1987)]. It was found that the scattering of phonons by N_2 vacancy is stronger than that by N_1 vacancy, which indicates that the N_2 vacancy has a greater negative effect on the thermal conductivity of MoSi_2N_4 .

In order to further demonstrate the influence of N-vacancy type and concentration on the phonon scattering and thermal conductivity, the parameters were obtained from first-principles calculations at 300 K. Subsequently, we plotted the phonon scattering rate as a function of phonon frequency (Fig. R6a) with a N-vacancy density of 0.01%. For the low frequency phonon scattering, the Umklapp phonon-phonon scattering is dominant. The phonon scattering by the change of force constant caused by N_2 vacancy plays a major role in the high frequency phonon scattering, while the N_1 vacancy has relatively little influence on the phonon scattering. Additionally, we plotted the normalized thermal conductivity of MoSi_2N_4 as a function of N-vacancy density. As shown in Fig. R6b, as the density of N_1 and N_2 vacancy increased, the normalized thermal conductivity decreased rapidly first and then decreased slowly. It is worth noting that N_2 vacancy caused a more pronounced decrease in thermal conductivity compared with N_1 vacancy.

Fig. R6 | Phonon scattering rate and thermal conductivity of defective MoSi_2N_4 at 300 K. (a) The phonon scattering rate as a function of phonon frequency with a N-

vacancy density of 0.01%. The inset shows the side view of the atomic structure of monolayer MoSi₂N₄. (b) The ratio of the thermal conductivity of defective MoSi₂N₄ to that of pristine one at room temperature as a function of N-vacancy density.

In summary, the phonon scattering is enhanced and the thermal conductivity is decreased with the increase of N-vacancy density in monolayer MoSi₂N₄. Moreover, N₂ vacancy has a greater negative effect on thermal conductivity because the phonon scattering of N₂ vacancy is stronger than that of N₁ vacancy. We have added Figure R6 as Supplementary Fig. 15 and related discussions in the revised manuscript. In addition, the detailed calculations and theoretical analyses have been added as Supplementary Note 3. The calculation method is shown below.

The Matthiessen's rule was used to combine the different phonon scattering mechanisms, and therefore

$$\tau^{-1} = \tau_U^{-1} + \tau_B^{-1} + \tau_V^{-1} + \tau_A^{-1} \quad \text{Eq. (R9)}$$

Where τ^{-1} is the combined phonon scattering rate, τ_U^{-1} is the Umklapp phonon-phonon scattering rate, τ_B^{-1} is the phonon-boundary scattering rate, τ_V^{-1} is the phonon-vacancy scattering rate, and τ_A^{-1} is the phonon scattering rate caused by the change of force constant. For the suspended single crystal samples, we mainly focused on τ_U^{-1} , τ_V^{-1} and τ_A^{-1} .

Slack G. A. and Galginaitis S. suggested the following form for the Umklapp phonon-phonon scattering rate [*Phys. Rev.* **133**, A253-A268 (1964)]:

$$\tau_U^{-1} = p\omega^2 \left(\frac{T}{\theta}\right) \exp\left(-\frac{\theta}{3T}\right) = C_U\omega^2 \quad \text{Eq. (R10)}$$

Where p is a constant, ω , T and θ are the circular frequency, temperature and Debye temperature, respectively. And C_U is the aggregation of other parameters except for ω .

Perturbation theory of Klemens P. G. described the phonon-vacancy scattering rate as [*Carbon* **32**, 735-741 (1994)]:

$$\tau_V^{-1} = x \left(\frac{\Delta M}{M}\right)^2 \frac{\pi \omega^2 g(\omega)}{2 G_N} \quad \text{Eq. (R11)}$$

Where x is the density of vacancies, $\frac{\Delta M}{M}$ is the effective mass, which is equal to $-\frac{M_a}{M} - 2$, M_a and M are the mass of vacancy atom and the average mass per atom,

respectively, $g(\omega)$ is the phonon density of states, and G_N is the number of atoms in the crystal.

Based on the bond-order theory for phonon-vacancy scattering [*J. Phys.: Condens. Matter* **14**, 7781-7795 (2002); *Nanotechnology* **16**, 1290-1293 (2005); *Prog. Solid State Chem.* **35**, 1-159 (2007); *Sci. Rep.* **4** 5085 (2014)], the scattering rate of phonons caused by the change of force constant was derived as:

$$\tau_A^{-1} = 4\pi z x \left\{ \left[\frac{1 + \exp[(12 - z)/8z]}{1 + \exp[(13 - z)/(8z - 8)]} \right]^{-(m+2)} - 1 \right\}^2 \frac{\omega^2 g(\omega)}{G_N} \quad \text{Eq. (R12)}$$

Where z is the effective coordination number, m is a parameter that represents the nature of bond, for compounds, m is around 4.

In monolayer MoSi₂N₄, the nitrogen atoms have two different lattice sites, and the inner N₁ atom and the surface N₂ atom have effective coordination numbers of 4 and 3, respectively. Thus, N₁ and N₂ vacancies have the same τ_V^{-1} but different τ_A^{-1} . Based on Eq. (R11) and (R12), τ_A^{-1} is 1.55 and 9.36 times larger than τ_V^{-1} for N₁ and N₂ vacancies, respectively.

3) The theoretical thermal conductivity is calculated by solving the phonon Boltzmann transport equation, as implemented in the ShengBTE package. The results for the convergence of the thermal conductivity with respect to the q-grid has to be presented.

Response: We thank the reviewer very much for the constructive suggestions.

We have tested the convergence of the thermal conductivities of MoSi₂N₄ and Si₂N₂ by using 5 - 10 atoms cutoff distances and 81 × 81 × 1 – 131 × 131 × 1 q-grid. As shown in Fig. R2, the thermal conductivity of MoSi₂N₄ at 300 K converges to 492.7 W·m⁻¹·K⁻¹ at 8 atoms cutoff distance and 111 × 111 × 1 q-grid, and the thermal conductivity of Si₂N₂ at 300 K converges to 819.5 W·m⁻¹·K⁻¹ at 9 atoms cutoff distance and 121 × 121 × 1 q-grid. These results indicate that the intrinsic thermal conductivity was underestimated slightly for MoSi₂N₄ in our original manuscript. The ultrahigh thermal conductivity of Si₂N₂ suggests that the outer Si-N layer in MoSi₂N₄ plays an important role for thermal conduction.

We have updated the related data and added Figure R2 as Supplementary Fig. 17 in the revised manuscript.

4) There is an incorrect description of “The third-order harmonic (3rd) IFCs ...” in Line 361-363. Harmonic vibration is only referred to second-order interatomic force constants. The term “harmonic” should be corrected to “anharmonic”.

Response: We thank the reviewer very much for pointing out this spelling mistake. We have changed “harmonic” to “anharmonic” in the revised manuscript.

5) Besides, in the 3rd IFCs calculations, the authors only considered the interactions up to 5 Å, is this cutoff distance accurate enough?

Response: We thank the reviewer very much for the insightful comments.

We are sorry that we mistyped the unit of the cutoff distance. In fact, the cutoff distance is 5 atoms rather than 5 Å. We performed convergence tests with more cutoff distances and q-grid values, which indicate that the cutoff distance of 8 atoms and the q-grid of $111 \times 111 \times 1$ are the convergence standards for MoSi_2N_4 , as shown in Fig R2. The related data have been updated in our revised manuscript.

Reviewer #3:

The manuscript written by Chengjian He et al. had measured thermal conductivity of $\sim 173 \text{ W}\cdot\text{m}^{-1}\cdot\text{K}^{-1}$ at room temperature for monolayer MoSi_2N_4 grown by chemical vapor deposition. Noncontact optothermal Raman technique and First-principles calculations reveal that such unusually high thermal conductivity benefits from the high Debye temperature and small Grüneisen parameter of MoSi_2N_4 . They are strongly dependent on the high Young's modulus induced by the outmost Si-N layers, which is interesting. However, there are some questions need to be answered.

Response: We thank the reviewer very much for the positive comments and constructive suggestions, which have helped us greatly improve the quality and readability of our manuscript.

1. Among various vdW layered 2D semiconductor, why do the authors choose MoSi_2N_4 ? Does it have any specific superior performance parameters that have been experimentally verified? Or some potentially huge application? The author doesn't seem to explain it very well.

Response: We thank the reviewer very much for the kind comments.

MoSi_2N_4 is a newly emerging and artificial 2D semiconductor, which has no known natural 3D counterparts and is grown by CVD. Many interesting properties of monolayer MoSi_2N_4 have been predicted benefiting from its unique sandwich structure, including high electron/hole mobilities, excellent thermal conductivity, valley pseudospin, quantum magneto-transport, piezoelectricity, sliding ferroelectricity, strong exciton-phonon coupling, second harmonic generation, photocatalysis and tunable Schottky barrier height. For instance, the electron/hole mobilities and thermal conductivity of monolayer MoSi_2N_4 were predicted to be up to 6 times higher than those of typical 2D semiconductor, monolayer MoS_2 . Another example is sliding ferroelectricity, where the vertical polarization of bilayer MoSi_2N_4 was predicted to be 10 times higher than that of WTe_2 . Experimentally, we have shown that monolayer MoSi_2N_4 have much higher Young's modulus, breaking strength and optical transmittance than those of monolayer MoS_2 [*Science* **369**, 670-674 (2020)]. Very recently, we found that monolayer MoSi_2N_4 has excellent nonlinear optical properties,

with second harmonic generation intensity about 10 times higher than that of monolayer MoS₂. These remarkable properties make monolayer MoSi₂N₄ a great potential for the applications in next-generation electronics and optoelectronics.

We have added more introduction about the advantages of MoSi₂N₄ over other 2D semiconductors in our revised manuscript.

2. “the known 2D semiconductors” in Abstract is a very large range, and the use of words like “highest” is discouraged unless there is solid evidence and a complete comparison. Kindly remove such words.

Response: We thank the reviewer very much for the kind suggestion. We have removed the sentence “which is the highest among the known 2D semiconductors” in the revised manuscript.

3. The references are only 16/65 in the past three years, and some references are too old. Proposed update.

Response: We thank the reviewer very much for the kind suggestions. We have carefully investigated the related papers reported in the past three years and accordingly updated the references in the revised manuscript.

4. Voigt function fitting and Linear fitting inevitably have errors, so the final thermal conductivity value of monolayer MoSi₂N₄ is a certain range, the more scientific form should be $A \pm B \text{ W} \cdot \text{m}^{-1} \cdot \text{K}^{-1}$.

Response: We thank the reviewer very much for the constructive suggestion.

As mentioned by the reviewer, both Voigt function fitting and linear fitting inevitably have errors. Taking into account the errors of data fittings, the thermal conductivity of monolayer MoSi₂N₄ was recalculated as $173.03 \pm 4.04 \text{ W} \cdot \text{m}^{-1} \cdot \text{K}^{-1}$ according to the Error Propagation Formula in Statistics.

We have updated Fig. 3h and k with error bars and given the new thermal conductivity value of monolayer MoSi₂N₄ in the revised manuscript.

5. As mentioned by the authors, the diameter of the hole also affects the heat dissipation of MoSi₂N₄. How to determine if a hole with a diameter of 6 μm is large enough for heat flow transfer from the center to the edge of monolayer MoSi₂N₄? And the substrate is a periodic through-hole arrays of 3, 4, 5, 6 μm in diameter. Is there any experimental

data with smaller holes to support the author's judgment?

Response: We thank the reviewer very much for the insightful comments.

In order to ensure enough accuracy of thermal conductivity extraction, the temperature of suspended monolayer MoSi₂N₄ near the hole edge needs to reach ambient temperature [ACS Nano 8, 986-993 (2014); J. Phys. Chem. C 125, 16129-16135 (2021)]. COMSOL simulation is a powerful method to simulate the spatial distribution of temperature [ACS Appl. Mater. Interfaces 9, 43013-43020 (2017)]. In our experiments, the maximum incident laser power was 3 mW, and the laser spot size was 1.18 μm. Under such experimental conditions, the temperature of suspended monolayer MoSi₂N₄ near the hole edge for different hole size was simulated. As shown in Figure R7, the temperature near the edge decreases with increasing the hole size and it reaches 300.06 K (ambient temperature) for a 6-μm-diametered hole (temperature distribution profile see in Fig. R4), indicating that 6-μm-diametered hole is large enough to ensure the accuracy of thermal conductivity extraction.

Figure R7 | Hole-diameter dependent temperature of suspended monolayer MoSi₂N₄ near the hole edge.

According to the reviewer's suggestion, we further performed thermal conductivity measurements on suspended monolayer MoSi₂N₄ with varying hole diameters (3, 4, 5,

6 μm). As shown in Fig. R8 and Table R2, the measured thermal conductivity increases first and then tends to converge when the hole diameter increases from 3 μm to 6 μm , confirming that a hole with a diameter of 6 μm is large enough for heat flow transfer from the center to the edge of monolayer MoSi_2N_4 . The similar phenomenon has been observed in TMDC materials [*Phys. Rev. Appl.* **13**, 034059 (2020)]. This is because phonons with long mean free paths (MFPs) are easily scattered by hole edge, which makes the measured thermal conductivity lower for smaller holes. As the hole diameter increases, low frequency phonons with longer MFPs are excited [*Phys. Rev. Appl.* **13**, 034059 (2020)], making the measured value closer to the intrinsic ones.

We have added Fig. R7, Fig. R8, Table R2 and the detailed simulations as Supplementary Fig. 7, Supplementary Fig. 10, Supplementary Table 2 and Supplementary Note 1, respectively, in the revised manuscript.

Figure R8 | The measured thermal conductivities of monolayer MoSi_2N_4 suspended on holes with the diameter of 3, 4, 5, and 6 μm . The red curve represents a log function fitting.

Table R2 | The $\chi / (\delta\omega/\delta p)$ and the derived thermal conductivities of monolayer MoSi_2N_4 suspended on holes with the diameter of 3, 4, 5 and 6 μm .

Hole diameter (μm)	$\chi / (\delta\omega / \delta p)$ ($\text{mW}\cdot\text{K}^{-1}$)	Thermal conductivity ($\text{W}\cdot\text{m}^{-1}\cdot\text{K}^{-1}$)
3	0.016426 ± 0.001706	134.87 ± 14.00
4	0.019141 ± 0.001042	157.16 ± 8.55
5	0.020670 ± 0.002118	169.71 ± 17.39
6	0.021649 ± 0.000295	177.75 ± 2.42

6. Is there a more detailed and physical explanation for the Raman peak redshift of MoSi_2N_4 suspended and supported on Au/SiO₂/Si substrates in Figure 3e?

Response: We thank the reviewer very much for the valuable suggestion.

The Raman peaks of a material stem from the vibrations or rotations of its chemical bonds. The position of Raman peak is related to the frequency of lattice vibration or rotation mode [*J. Raman Spectrosc.* **11**, 346-348 (1981); *Small* **9**, 2857-2861 (2013)], which is sensitive to the lattice deformation. Generally, a tensile and compressive strain in materials will cause a redshift and blueshift of Raman peaks, respectively, ascribed to the softening and hardening of chemical bonds [*2D Materials* **2** 024009 (2015); *J. Alloys Compd.* **961** 170998 (2023)]. Different from the MoSi_2N_4 supported on Au/SiO₂/Si substrates, there would be a tensile stress in the suspended MoSi_2N_4 induced by its own gravity. The corresponding tensile strain results in the Raman peak redshift.

We have added the above discussions in the revised manuscript.

7. Why might point defects reduce the thermal conduction of Si-N layers? Are other material preparation methods, such as mechanical stripping, expected to improve the situation?

Response: We thank the reviewer very much for the insightful comment and constructive suggestion.

The effects of point defects on thermal conductivity have been widely reported for various 2D materials [*Phys. Lett. A* **376**, 3668-3672 (2012); *AIP Adv.* **7**, 105110 (2017); *Phys. Rev. Mater.* **4**, 014004 (2020)]. Previous experimental and theoretical results have demonstrated that phonon scattering is enhanced by defects, and thus thermal conductivity decreases with the increase of defect density. To reveal how point defects

in MoSi₂N₄ reduce the thermal conduction of Si-N layers, we conducted DFT calculations. The N vacancies at different positions were investigated, where the inner and outer nitrogen vacancy was named as N₁ and N₂ vacancy, respectively. We plotted the phonon scattering rate as a function of phonon frequency with a vacancy density of 0.01% (Fig. R6a). For the low frequency phonon scattering, the Umklapp phonon-phonon scattering is dominant. The phonon scattering caused by N₂ vacancy plays a major role in the high frequency phonon scattering, while N₁ vacancy has a relatively little impact on the phonon scattering. Therefore, N₂ vacancies in outer Si-N layer significantly reduce the thermal conduction of Si-N layers. Additionally, we plotted the normalized thermal conductivity of MoSi₂N₄ as a function of N-vacancy density (Fig. R6b). As the density of N₁ and N₂ vacancy increased, the normalized thermal conductivity decreased rapidly first and then decreased slowly. Moreover, N₂ vacancy caused a more pronounced decrease in thermal conductivity compared with N₁ vacancy. We have added Fig. R6 as Supplementary Fig. 15.

As the reviewer mentioned, mechanical stripping is indeed an ideal method for the fabrication of high-quality 2D layered materials. However, different from the other 2D layered materials, MoSi₂N₄ is an artificial 2D layered material without existing 3D bulk counterpart [*Science* **369**, 670-674 (2020)]. Although multilayer MoSi₂N₄ crystals with a small lateral size (less than 5 μm) have been grown by CVD method, it is not suitable for mechanical stripping. Once high-quality bulk counterparts are achieved in the future, mechanical stripping can be used to improve the current situation.

8. Why the wrinkles in MoSi₂N₄ can enhance phonon scattering? And is “enhance phonon scattering” necessarily related to “reduce the thermal conductivity”?

Response: We thank the reviewer very much for the insightful comments.

The influence of wrinkles on phonon transport has been extensively investigated for various 2D materials. Wrinkle, as a form of out-of-plane torsional deformation [*Mater. Today Commun.* **22** 100706 (2020); *J. Appl. Phys.* **129**, 233101 (2021)], can result in strong phonon localizations which are concentrated on the joint regions between crests and troughs, as well as enhanced phonon-boundary scattering as evidenced by vibrational density of states attenuation in the low frequency region [*J. Phys. Chem. C*

120, 23807-23812 (2016); *RSC Adv.* **7**, 54734-54740 (2017)]. In addition, the undulated heat transfer path can enhance phonon scattering [*Mol. Simul.* **41**, 231-236 (2015)]. It has been predicted that phonon scattering is enhanced with increasing the number and amplitude of wrinkles [*Mater. Today Commun.* **22** 100706 (2020)]. In MoSi₂N₄, the thermal conduction is mainly dependent on phonon transport. Thus, the wrinkles-enhanced phonon scattering can reduce the thermal conductivity of MoSi₂N₄. The same phenomenon has been found in graphene [*J. Phys. Chem. C* **120**, 23807-23812 (2016); *J. Appl. Phys.* **129**, 233101 (2021); *Nanotechnology* **23** 365701 (2012)].

We have added the above discussions in the revised manuscript.

9. It is suggested the authors to check their manuscript carefully and thoroughly to avoid some minor typical mistakes and mistypes. Such as in Figure 3k, what is “laser powder-dependent”?

Response: We thank the reviewer very much for the kind reminder and correction.

“Powder” in “laser powder-dependent” is a spelling mistake. We have changed “powder” to “power” in the revised manuscript and checked our manuscript carefully to avoid spelling mistakes.

10. In Figure 3k, what’s the difference if we change the wavelength of the excitation laser?

Response: We thank the reviewer very much for the constructive suggestion.

According to the mechanism of optothermal Raman measurements, the thermal conductivity value of 2D materials is derived from Eq. (1) - (3) as shown in our manuscript. In these equations, the wavelength of the excitation laser is not involved, and thus the measured thermal conductivity is independent on the wavelength of the excitation laser. However, the laser wavelength can affect the accuracy of thermal conductivity measurements since precise identification of the Raman peak position is required for optothermal Raman measurements. For example, the first report of thermal conductivity measurement on graphene using optothermal Raman technique demonstrated that 325-nm-wavelength laser could not provide clear Raman signatures, so the authors used 488-nm laser for the measurements [*Nano Lett.* **8**, 902-907 (2008)]. In another report, the authors found that three different lasers (325, 488, 633 nm) can

stimulate Raman signals of FePS₃ and MnPS₃, but they used 633-nm laser for optothermal Raman measurements because it can stimulate intense and narrow A_g mode with most pronounced temperature dependence [*ACS Nano* **14**, 2424-2435 (2020)].

According to the reviewer's suggestion, we have conducted optothermal Raman measurements on monolayer MoSi₂N₄ with excitation lasers of 488 nm, 532 nm, 633 nm and 785 nm, including samples supported on SiO₂/Si substrate and suspended on through holes. As shown in Fig. R9, monolayer MoSi₂N₄ shows no Raman peaks under 633 nm and 785 nm lasers. Although a weak Raman signal can be observed under 488 nm laser, it's too weak to precisely extract the Raman peak position. In contrast, a pronounced Raman signal can be observed under 532 nm excitation laser. This is because B exciton energy in monolayer MoSi₂N₄ corresponds to an energy level of 2.35 eV, which closely matches the energy of 532 nm laser (~2.33 eV) [*Science* **369**, 670-674 (2020)]. Given that thermal conductivity measurement by optothermal Raman technique requires strong and sensitive Raman signal response [*ACS Nano* **14**, 2424-2435 (2020)] to reduce the errors, we used 532 nm laser for the measurements in our experiments. We have added Fig. R9 as Supplementary Fig. 2 in the revised manuscript.

Figure R9 | Raman spectra of monolayer MoSi₂N₄ supported on SiO₂/Si substrate (a) and suspended on through holes (b) under excitation lasers with different wavelengths of 532 nm, 785 nm, 633 nm and 488 nm.

REVIEWERS' COMMENTS

Reviewer #1 (Remarks to the Author):

The manuscript has been convincingly revised. I have no further comments.

Reviewer #2 (Remarks to the Author):

The authors have fully considered the reviewers' comments, carefully revised the manuscript, and added additional experimental and simulation results to remarkably improve the quality of this work. Therefore, the reviewer recommends the publication of this work in the current form.

Reviewer #3 (Remarks to the Author):

The authors have made the revision. I have no other comments.

Response to reviewers' comments

Reviewer #1 (Remarks to the Author):

The manuscript has been convincingly revised. I have no further comments.

Response: We thank the reviewer very much for the positive comments.

Reviewer #2 (Remarks to the Author):

The authors have fully considered the reviewers' comments, carefully revised the manuscript, and added additional experimental and simulation results to remarkably improve the quality of this work. Therefore, the reviewer recommends the publication of this work in the current form.

Response: We thank the reviewer very much for the positive comments.

Reviewer #3 (Remarks to the Author):

The authors have made the revision. I have no other comments.

Response: We thank the reviewer very much for the positive comments.